# Energy systems in scenarios at net-zero CO$_2$ emissions

Julianne DeAngelo [1✉], Inês Azevedo[2], John Bistline[3], Leon Clarke[4], Gunnar Luderer [5], Edward Byers [6] & Steven J. Davis [1,7]

Achieving net-zero CO$_2$ emissions has become the explicitgoal of many climate-energy policies around the world. Although many studies have assessed net-zero emissions pathways, the common features and tradeoffs of energy systems across global scenarios at the point of net-zero CO$_2$ emissions have not yet been evaluated. Here, we examine the energy systems of 177 net-zero scenarios and discuss their long-term technological and regional characteristics in the context of current energy policies. We find that, on average, renewable energy sources account for 60% of primary energy at net-zero (compared to ~14% today), with slightly less than half of that renewable energy derived from biomass. Meanwhile, electricity makes up approximately half of final energy consumed (compared to ~20% today), highlighting the extent to which solid, liquid, and gaseous fuels remain prevalent in the scenarios even when emissions reach net-zero. Finally, residual emissions and offsetting negative emissions are not evenly distributed across world regions, which may have important implications for negotiations on burden-sharing, human development, and equity.

[1] Department of Earth System Science, University of California, Irvine, Irvine, CA, USA. [2] Department of Earth System Science, Stanford University, Stanford, CA, USA. [3] Electric Power Research Institute, Palo Alto, CA, USA. [4] School of Public Policy, University of Maryland, College Park, MD, USA. [5] Potsdam Institute for Climate Impact Research, Potsdam, Germany. [6] Energy Program, International Institute for Applied Systems Analysis, Laxenburg, Austria. [7] Department of Civil and Environmental Engineering, University of California, Irvine, Irvine, CA, USA. ✉email: deangelj@uci.edu

Limiting global mean temperature increase to 2 °C or even 1.5 °C relative to the preindustrial era[1] requires that global annual $CO_2$ emissions are net-zero or net-negative by the end of this century, and perhaps as soon as 2050[2–5]. In the broader context of climate stabilization, the magnitude of global temperature increase is directly proportional to cumulative $CO_2$ emissions, such that adding any amount of $CO_2$ to the atmosphere will increase future amounts of warming[2,6]. For these reasons, and because it is a clear and absolute target, achieving net-zero emissions is increasingly a goal of energy and emissions policies around the world[3,7–10]. Central to meeting this goal is a rapid and sweeping transformation of energy systems, including drastic reductions in the use of fossil fuels, substantial improvements in energy and materials efficiency, extensive electrification of energy end uses, and management of carbon[11–16]. Moreover, this transformation of energy systems must be reconciled with both sustainable development goals[17,18] and the considerable inertia of existing fossil energy infrastructure[19].

Given this context, energy analysts are increasingly exploring the challenges and opportunities for net-zero emissions energy systems[20], including detailed analyses of specific energy services and/or technologies[21–25]. A number of recent studies have examined the mitigation pathways of energy systems in integrated assessment model (IAM) scenarios that limit warming to below 1.5 °C[26–30], providing insight about possible transformations of the energy-economy-land system. However, the common features and tradeoffs of such scenarios at the point when global $CO_2$ emissions reach net-zero have yet to be systematically assessed. These characteristics at the point of net-zero $CO_2$ can inform policies that might take varying approaches – including potential approaches that are not represented by current scenario pathways – to reach the same goal of net-zero emissions.

Here, we analyze 177 IAM scenarios from the public 1.5 °C Scenario Database (the SR1.5 database)[31,32] in which global sources and sinks (including land use and agriculture) reach net-zero $CO_2$ emissions by 2100 (see Supplementary Table 1). Details of our processing and analytic approach are described in the Methods section. In summary, we assess global and regional energy use, energy sources, residual emissions, electrification, and climate policy among the scenarios, finding robust features that span multiple IAMs[33]. For example, renewable sources represent roughly 60% of primary energy at the point when they reach net-zero $CO_2$ emissions—and often more than half of such renewable energy is provided by biomass. However, it is important to note that the scenario ensemble is not a representative sample that can be used to infer likelihood; individual scenarios are equally plausible given model constraints.

## Results

**Energy use and timing of net-zero.** Figure 1 shows the relationships among global energy and socioeconomic variables in the year of global net-zero emissions, broken out by the level of projected global warming. These categories include overshoot scenarios that return to the specified amount of warming by the end-of-century (see Methods). Among the 177 net-zero scenarios, those that avoid mean end-of-century warming of 1.5 °C (blue points) tend to have lower levels of global energy use (t-statistic = 9.2, $p < 0.001$) and less GDP per capita (t-statistic = 8.6, $p < 0.001$): of the 77 1.5 °C scenarios, GDP per capita is < $40,000 per person per year in 91% (median $27,914, range $20,103–$58,506) and total final energy use is <500 EJ in 69% (median 439 EJ, range 227–646; Fig. 1a). In contrast, energy use and GDP per capita are substantially higher in scenarios that achieve net-zero emissions but exceed 1.5 °C (green and orange points): of the 100 2 °C and >2 °C scenarios, GDP per capita

is < $40,000 per person per year in only 43% (median $43,642, range $20,299–$116,666) and total final energy use is <500 EJ in 24% (median 580 EJ, range 345–857; Fig. 1a). Although this may reflect reduced energy use and economic activity in scenarios with the most ambitious mitigation, it is also related to when net-zero emissions occur in these scenarios. Supplementary Fig. 1 supports this idea by showing that warming level is not strongly related to the levels of energy use and GDP ultimately reached in net-zero scenarios. Figure 1b shows that the warmer scenarios achieve net-zero emissions in progressively later years (median for all scenarios = 2064, range 2037–2100), because the additional time for the economy and energy system to grow in these scenarios leads to higher cumulative $CO_2$ emissions (and therefore higher levels of subsequent warming). Supplementary Figs. 2 and 3 support this idea that more ambitious scenarios achieve lower levels of warming via faster energy system transformations. However, in contrast to the timing of net-zero, the timing of peak emissions is consistent across the scenarios (and essentially immediate): emissions peak in 2017 (range 2014–2027) for 1.5 °C scenarios, in 2019 (range 2011–2029) for 2 °C scenarios, and in 2022 (range 2010–2036) for >2 °C scenarios (Fig. 1b). Although many scenarios show emissions peaking prior to 2019 (which did not occur), the regional, socio-economic, and technological representations that prevail when these scenarios achieve net-zero emissions may nonetheless provide valuable insights for net-zero emissions policies.

**Energy sources.** The use and sources of renewable energy in net-zero scenarios vary considerably, with no obvious relationship to the level of warming (Fig. 1c). Although the median share of primary energy derived from renewable sources (including biomass, solar, wind, hydroelectricity, and geothermal, using the direct equivalent method[34]) is ~60% regardless of warming level, in some cases it is as little as 25% and reaches 80% in a few others (Fig. 1c). Similarly, the median share of these renewables that are not biomass is ~55% regardless of warming level, but ranges from 20–89% (Fig. 1c). Supplementary Fig. 4 further decomposes the sources of primary energy in net-zero scenarios, showing, for example, that the largest share of primary energy from nuclear is 23%, with nuclear more often contributing a small share of energy (median share across all scenarios is 4.8%, range 0–23.4%). Moreover, the share of primary energy from fossil fuels (coal, oil, and natural gas) in net-zero scenarios with and without carbon capture ranges from 3–64%, with a median share across all scenarios of 33% (Supplementary Fig. 4). By definition, in net-zero scenarios, any residual emissions to the atmosphere from the use of fossil fuels are offset by negative emissions strategies.

**Residual emissions and electrification.** The scale of residual emissions, i.e. emissions that are counter-balanced by equivalent carbon sequestration, is important to consider given many feasibility concerns about negative emissions technologies[33,35]. Figure 1d shows that the emissions intensity of final energy may remain quite high in net-zero scenarios (e.g., >30 Mt CO2/EJ compared to the current level of ~80 Mt CO2/EJ). This residual emissions intensity is insensitive to the warming level or the energy intensity of the global economy (although lower warming scenarios do tend to have lower energy intensities based on median values by warming group; Fig. 1d). Given that the points depicted in Fig. 1d are globally net-zero, the residual emissions are entirely offset by negative emissions.

Complementing the common assertion that everything must be electrified[36,37], the scenario set indicates that reducing final energy use is also an important determinant for achieving 1.5 or 2 °C. Electricity accounts for 35–80% of final energy across the

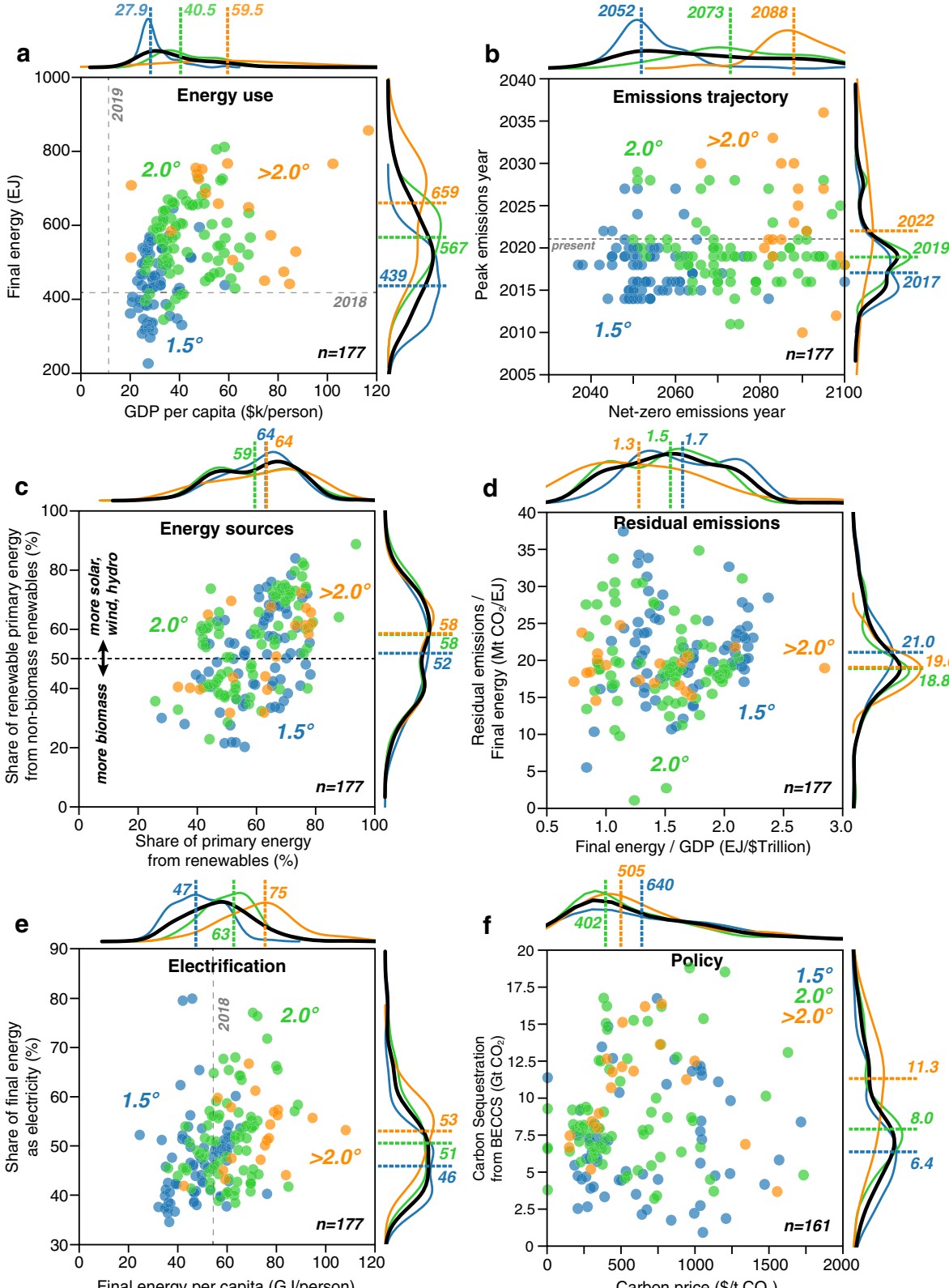

**Fig. 1 Energy system parameters in global net-zero CO2 emissions scenarios.** Scenarios that reach net-zero emissions show differences in energy use (**a**), emissions trajectory (**b**), energy sources (**c**), residual emissions (**d**), electrification (**e**), and policy (**f**), particularly with respect to warming levels (blue = <1.5 °C, green = <2.0 °C, orange = >2.0 °C). Points represent individual scenarios, with a frequency of scenarios shown along each axis for each warming level (colors corresponding to warming levels) and for all scenarios (black). Colored dashed lines and values indicate medians for warming groups, with colors corresponding to warming groups. Gray dashed lines indicate reference values for the year shown in gray.

range of net-zero scenarios, but is <70% in most >2 °C scenarios (Fig. 1e). Even though electrification is a useful mechanism for decarbonization, warmer scenarios tend to exhibit slightly higher levels of electrification at the timing of net-zero: median shares of 1.5 °C, 2 °C and >2 °C scenarios are 46% (range 35–80%), 51% (range 38–77%), and 53% (range 42–67%), respectively, perhaps because they afford greater time for end-uses to transition (Fig. 1e). This transition-time effect on the amount of electrification is supported by Supplementary Fig. 3, which shows that scenarios that are later in reaching net-zero tend to compensate with higher amounts of electrification (Supplementary Fig. 3e). Warming amount is also correlated to both net-zero year ($r = 0.73$, $p < 0.001$; Fig. 1b) and electrification ($r = 0.27$, $p < 0.001$) in the Fig. 1 global scenarios, which further supports the idea that warmer scenarios have slightly higher amounts of electrification because they reach net-zero emissions later, thus allowing more time for end-uses to transition and for costs to decline. However, these are subtle distinctions in comparison to the differences in per capita final energy use, where median shares in 1.5 °C, 2 °C and >2 °C scenarios increase from 47 to 63 to 75 GJ per person, respectively. For comparison, in 2019 the average American, EU, and Chinese citizen used approximately 202, 93, and 63 GJ, respectively. Thus, keeping final energy low is clearly important to meet 1.5 °C, while there is more flexibility in the level of electrification that is required.

**Negative emissions and policy.** The prevailing carbon prices in net-zero scenarios—a proxy for global climate policies—range from zero to > $1000/t $CO_2$, yet with no clear relationship to either warming level or the amount of carbon sequestration through bioenergy with carbon capture and storage (BECCS) (Fig. 1f; note that 16 scenarios with prices > $2000/t $CO_2$ are not shown). It is important to note that carbon prices in the majority of SR1.5 scenarios are endogenous "shadow" carbon prices that reflect the marginal cost of abatement, and thus do not directly reflect the impact of explicit (exogenous) carbon pricing such as a carbon tax or cap-and-trade system[33,38,39]. Only 23 of the 177 scenarios we analyze here include exogenous carbon pricing. The relationship between BECCS and carbon price should therefore be interpreted as the impact of marginal abatement cost on BECCS deployment. The lack of a clear relationship between the two does not necessarily mean that marginal abatement cost is inconsequential for the magnitude of negative emissions, but rather indicates that other dynamics relating to technology availability and costs may be the main drivers of BECCS deployment. Additionally, the median amount of carbon sequestration from BECCS increases in 1.5 °C, 2 °C and >2 °C scenarios, from 6.4 (range 0–16.7) to 8.0 (range 0–18.8) to 11.3 (range 3.7-16.4) Gt $CO_2$, respectively (Fig. 1f), indicating that warmer scenarios must rely on greater amounts negative emissions technologies to reach net-zero emissions.

**Regional energy use, energy sources, and electrification.** Figure 2 shows regional differences in energy and emissions among net-zero scenarios (in the year in which global $CO_2$ emissions are net-zero). In some cases, these differences are substantial and systematic. For example, Fig. 2a shows that when global emissions are net-zero, total final energy consumption is typically greatest in Asia (blue points) and the OECD and EU countries (e.g., the U.S., U.K., France, Germany, etc.; pink points)—in some cases more than 3 times the energy use in the Middle East and Africa, Latin America, and Eastern Europe (including Russia; yellow, green and purple points, respectively). Regional differences in GDP per capita in the net-zero year are somewhat less dramatic, but projections in the OECD and EU region are often greatest (median of $67,944 per

person, range $47,534–$146,341), and projections in the Middle East and Africa are often lowest (median of $18,960 per person, range $6,263–$97,721; Fig. 2a).

As in the case of globally aggregated energy sources (Fig. 1c), the share of primary energy derived from renewables and different types of renewables are quite different across scenarios, with relatively little sensitivity to the region (Fig. 2b). An exception is Latin America (green points), which most scenarios show having both a higher share of primary energy from renewables (median 80%, range 33–98%) and a greater share of those renewables from biomass (median 58%, range 12–83%) than other regions (median shares of renewables 58–67%, and median share of renewables from biomass 35–45%).

Regional variations in electrification are also small (regions' median shares range from 43–52%), though final energy use per capita varies across regions in a pattern similar to GDP per capita (Fig. 2a and c; Supplementary Fig. 5). Despite lower GDP per capita, energy use per capita in Eastern Europe and Russia is similar to the OECD and EU region (median energy use of 105 and 112 GJ/person, respectively) — considerably greater than in the other three regions, where median energy use ranges from 36–61 GJ/person (Fig. 2c; note that Eastern Europe and Russia per capita final energy exceeded 200 GJ/person in 2 scenarios that are not shown).

**Regional Distribution of Residual and Negative Emissions.** Importantly, when global emissions are net-zero, emissions in many scenarios are still net-positive in some regions and (proportionately) net-negative in others. Figure 2d shows the regional balance of per capita residual emissions from energy and industry and per capita negative emissions from BECCS—i.e. net energy system emissions in the region (when points are compared to the dashed black line). These differences in residual (F-statistic = 141.6, $p < 0.001$) and negative emissions (F-statistic = 70.7, $p < 0.001$) across regions can be at least partially explained by differences in investment: Supplementary Fig. 6 shows that cumulative investment in non-fossil electricity supply up to the global net-zero year is correlated with regional electrification ($r = 0.55$, $p < 0.001$), negative emissions from BECCS ($r = 0.58$, $p < 0.001$), and residual emissions from energy and industry ($r = 0.86$, $p < 0.001$; Supplementary Fig. 6). The positive correlation between non-fossil electricity investment and both BECCS and residual emissions is likely due to BECCS primarily being used to offset residual emissions, such that scenarios with high amounts of BECCS also have high amounts of residual emissions at net-zero. Of course, investment is not the only cost-related driver of these regional characteristics, but it does appear to play a significant role in the smaller subset of scenarios that include investment output values. Residual emissions per capita tend to be greater in regions of Eastern Europe and Russia and the OECD and EU, with median values of 1.9 (range 0.1–5.2) and 1.8 (range 0.2–4.9) t $CO_2$/person, respectively (purple and pink points in Fig. 2d). However, these regions also have greater per capita negative emissions from BECCS than Asia and the Middle East and Africa regions, such that they are net-negative in nearly as many scenarios (40.1% and 49.4% for Eastern Europe and Russia and OECD+EU, respectively) as they are net-positive (59.9% and 50.6%, respectively). In contrast, Latin America's energy system is net-negative in 78.1% of the scenarios (green points) and the Middle East and Africa and Asia regions are net-negative in just 14.0% and 19.4%, respectively (orange and blue points). This supports recent research on regional and country-level negative emissions distributions in the context of regional net-zero emissions[40,41] and indicates that burden-sharing between currently less-developed regions may not be well-balanced in IAM outputs when global emissions reach net-zero. While there are many different approaches to defining a well-balanced mitigation effort[42], burden-

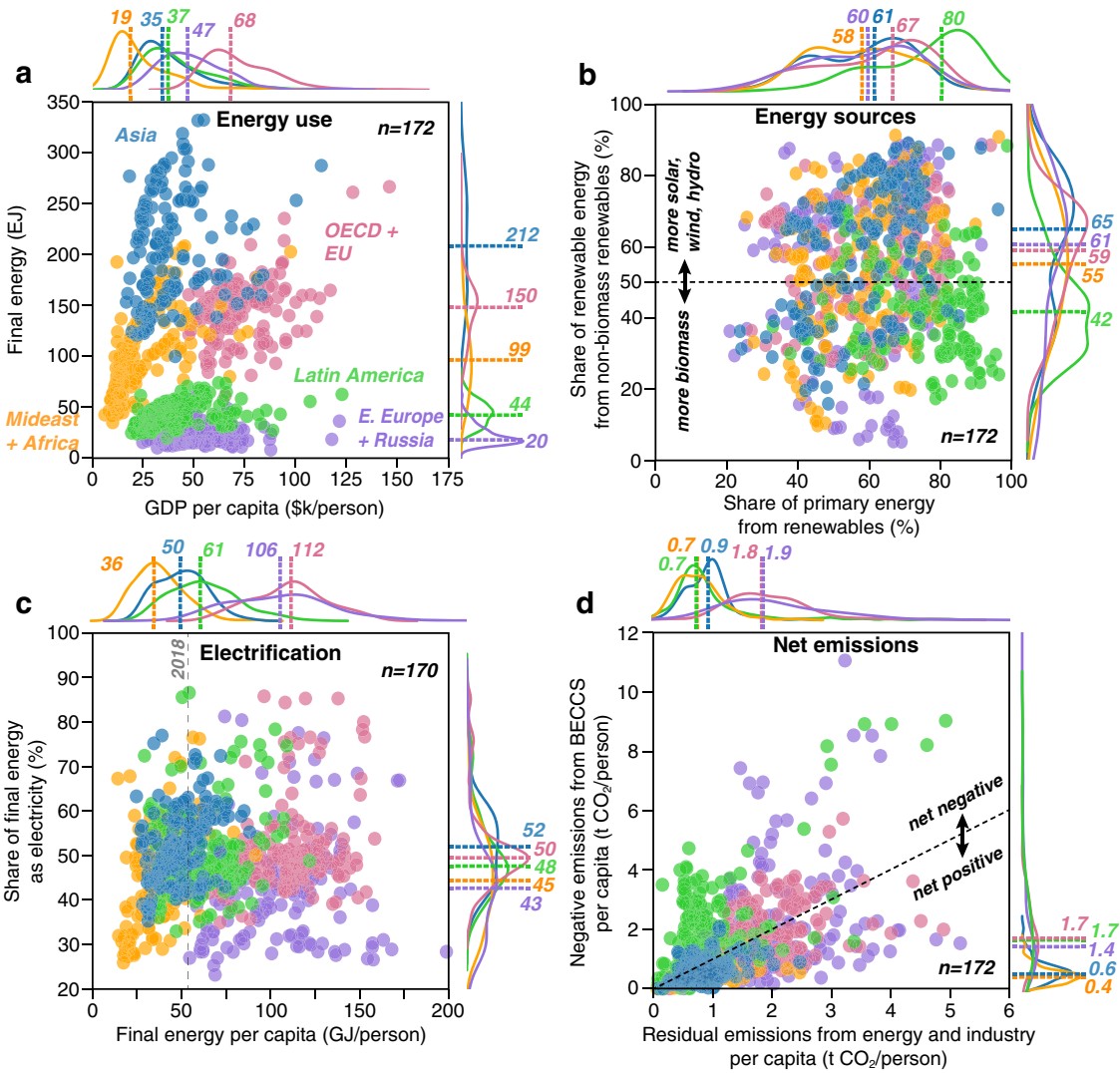

**Fig. 2 Characteristics of regional energy systems and emissions when global emissions reach net-zero.** Scenarios that reach net-zero emissions globally (*n* = 172 scenarios with all regions) show regional differences in energy use (**a**), energy sources (**b**), electrification (**c**), and net emissions (**d**). Points represent individual scenarios, with a frequency of scenarios shown along each axis for each region (Asia = blue, Latin America = green, Middle East +Africa = orange, OECD + EU countries = pink, and Eastern Europe+Russia = purple). Colored dashed lines and values indicate medians for each region. Gray dashed lines indicate global reference values for the year shown in gray.

sharing approaches that consider equity as a key component are vital for meeting sustainable development goals[43]. Analysis of the SR1.5 scenarios in the context of equitable emissions/negative emissions allocation and sustainable development warrants further research.

Figure 3a shows the global distributions of residual and negative emissions in net-zero scenarios, including both those explicitly tied to the energy system (i.e. residual emissions from energy and industrial processes and negative emissions from BECCS) and those related to agriculture and land use (including afforestation and reforestation), which are major sources of negative emissions in many IAMs[44]. The aggregate patterns are striking: in warmer scenarios, net emissions from agriculture and land use tend to be less negative, residual emissions are higher, and these trends must be compensated for by larger negative emissions from BECCS (Fig. 3a). In net-zero scenarios where warming is >2 °C, negative emissions from BECCS in the net-zero year are on average 10.5 Gt $CO_2$, and in no scenario <3.7 Gt (range 3.7–16.4; Fig. 3a). In contrast, there are some 1.5 °C and <2.0 °C scenarios in which there are no negative emissions from BECCS because more modest residual emissions are balanced by

larger negative emissions from land uses (excluding BECCS), such as afforestation (Table 1). The negative emissions from BECCS also decrease in more ambitious mitigation scenarios, with mean values of 8.7 (range 0–18.8) Gt $CO_2$ and 6.7 (range 0–16.7) Gt $CO_2$ for <2.0 °C and 1.5 °C scenarios, respectively (Fig. 3a; Table 1). Although residual emissions by end-use sector were not available for many of the scenarios we assessed, transportation was the dominant source of residual emissions in the 40 scenarios which report these details, followed by either the industry or residential and commercial sectors (see Supplementary Fig. 7).

Global averages conceal considerable regional heterogeneity of emissions in a net-zero world. Figure 3b shows that potential negative emissions from land use are largest in Latin America (on average −1.1 Gt $CO_2$ in the net-zero year, range −4.8 to 1.7 Gt), while Asia is projected to be by far the largest source of residual emissions (on average 3.8 Gt $CO_2$ in the net-zero year, range 0.3–10.3 Gt). Asia and the OECD and EU regions are also the largest sources of negative emissions from BECCS, with an average of 2.5 (range 0–8.7) and 2.4 (range 0–6.0) Gt negative $CO_2$ emissions in the net-zero year, respectively; Fig. 3b).

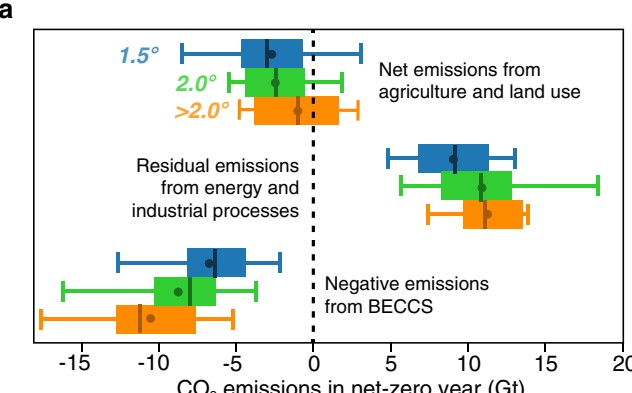

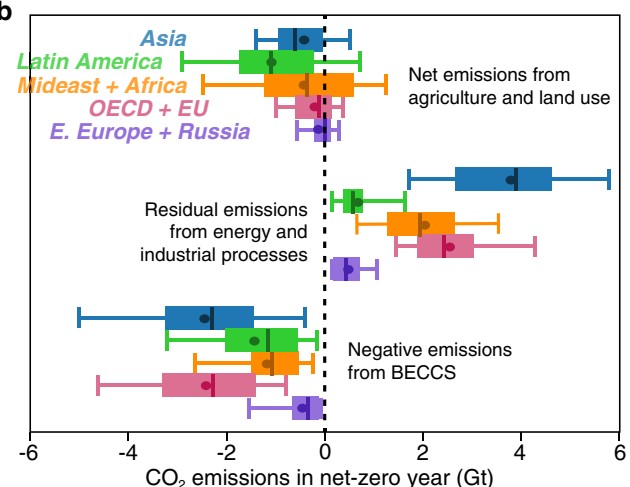

**Fig. 3 Residual and negative emissions when global emissions reach net-zero.** Residual and negative emissions in net-zero scenarios show global differences across different warming levels (**a**) and regions (**b**). In each case, the boxes show the range from 25th to 75th percentiles, whiskers show the 5th and 95th percentiles, and the lines and circles within the boxes denote the median and mean values, respectively.

**Relationships between scenario characteristics**. Figure 4 compares all 177 net-zero scenarios according to 6 global characteristics in the net-zero year: the share of final energy that is electricity, the share of primary energy derived from renewables, the share of renewable energy that is derived from non-biomass sources, energy conservation (i.e. the inverse of per capita energy demand), the magnitude of negative emissions from BECCS, and net land-use emissions. Each panel in Fig. 4 sorts all the scenarios (rows) according to one of these characteristics (columns), with scenario values shown as z-scores. Pairwise correlation coefficients (r) are also shown at the top of each column to quantitatively compare each set of parameters (Supplementary Fig. 8). Plotted this way, for example in (a), it is evident that those scenarios in which electricity accounts for a greater share of final energy also tend to be associated with greater shares of renewable energy ($r = 0.64$, $p < 0.05$) and non-biomass renewable energy ($r = 0.59$, $p < 0.05$), but less energy conservation (i.e. greater per capita energy use, $r = -0.35$, $p < 0.05$; Fig. 4a). Scenarios with greater shares of renewable energy tend to have higher shares of non-biomass renewable energy ($r = 0.50$, $p < 0.05$; Fig. 4b), while scenarios with greater amounts of energy conservation tend to have lower shares of non-biomass renewable energy, and vice versa ($r = -0.46$, $p < 0.05$; Fig. 4c and d). The relationship among these characteristics and the magnitude of negative emissions

from BECCS and/or net land-use emissions is less clear, and maybe more dependent on the IAM or specific scenario used in each case. Since the process-based IAMs considered here use cost-effectiveness analysis (CEA)[33], which minimizes the total mitigation costs of reaching a specified climate goal, all associations between output variables are essentially a reflection of what is cheapest. For example, in a scenario where substantial residual emissions remain at net-zero and are offset by correspondingly large amounts of negative emissions, reducing gross emissions to zero must have been more expensive than continuing to emit and offsetting with negative emissions. The most cost-effective outputs for scenarios are also based on the assumptions of individual models, including the availability and cost of technologies.

To further explore this relationship between negative emissions and other parameters, the underlying structure of the IAMs is important to consider: some of the SR1.5 models are partial equilibrium models (e.g., POLES ADVANCE) while others are general equilibrium (e.g. AIM-CGE 2.0 and 2.1) or hybrid models (e.g., MESSAGE-GLOBIOM 1.0) that link the two[31]. Additionally, certain scenarios have conditions that limit the amount or type of negative emissions technology used, such as EMF33_1.5C_limbio, which sets a limit of 100 EJ/year for the amount of bioenergy from BECCS, cellulosic fuels, and hydrogen[31]. Supplementary Fig. 9 shows the scenario ranges for residual emissions, non-biomass renewable energy share, and electrification for each model. These ranges demonstrate how the structure and assumptions of individual models affect the scenario outputs[45,46]: for example, GCAM scenarios tend to have systematically higher residual emissions and lower amounts of non-biomass renewable energy and electrification than those of other models (Supplementary Fig. 9). Such model differences are visible when comparing individual scenarios, but the output ranges tend to bemore sensitive to the scenario constraints than the models (Supplementary Fig. 10).

## Discussion

In addition to renewable and net-zero targets, "electrify everything" has become an explicit policy goal in a growing number of places[47], particularly regarding heating and cooking in the residential and commercial sectors[48,49] and light-duty transportation[50,51]. In contrast, in most net-zero scenarios, electricity accounts for less than half (median 48.5%) of final energy (Fig. 1e), including in the OECD and EU regions (Fig. 2c). Although electricity makes up a greater fraction of final energy in all net-zero scenarios than it does today (~20% today), in some regions and cases electricity remains less than 30% of final energy used (Fig. 2c). This emphasizes that IAMs project considerable ongoing use of solid, liquid and gaseous fuels in hard-to-electrify sectors (such as construction, agriculture, aviation and shipping) even when emissions are net-zero (Supplementary Fig. 11). In this context, lower levels of final energy use per capita is one of the more robust trends of 1.5 °C scenarios. Meanwhile, our finding that electricity is somewhat less prevalent at the net-zero point in scenarios with lower warming may reflect the additional time available for end uses to electrify in less ambitious (higher warming) scenarios (Fig. 1e and b).

Although the carbon intensity of final energy declines drastically in many net-zero scenarios compared to present (~80 Mt CO2/EJ; Fig. 1d), the absolute quantity of residual emissions remains substantial in many of the scenarios—as often as not >10 Gt CO2 globally in the net-zero year (Fig. 3). This translates into prodigious quantities of negative emissions required, with perhaps proportional social, techno-economic and biophysical challenges[15,35,52]. But we also find that both the residual emissions and the negative emissions required to offset them are not evenly distributed across world regions (Figs. 2d and 3b), which may have important implications for human development and equity[53]. In particular, net-zero scenarios

**Table 1 Parameter values and statistics for global scenarios in the net-zero year.**

| Warming Category | Statistic | Final energy as electricity (%) | Primary Energy, Fossil (%) | Primary Energy, Nuclear (%) | Primary Energy, Renewable (%) | Renewable Energy, Non-biomass Renewables (%) | Final energy per capita (GJ/person) | Carbon seq., BECCS (Mt CO2) | Net CO2 emissions, land use (Mt CO2) | Energy Intensity of GDP (EJ/$Trillion) | Net-Zero Year |
|---|---|---|---|---|---|---|---|---|---|---|---|
| <1.5 °C (n = 77) | Median | 46.1 | 32.6 | 4.8 | 63.5 | 51.9 | 47.4 | 6,356.8 | −2,967.6 | 1.7 | 2052 |
| | Mean | 46.7 | 34.0 | 5.8 | 60.2 | 51.8 | 48.3 | 6,722.3 | −2,696.9 | 1.7 | 2054 |
| | Min | 34.5 | 13.9 | 0 | 28.0 | 20.3 | 24.5 | 0 | −13,477.0 | 0.8 | 2037 |
| | Max | 79.9 | 61.9 | 15.7 | 78.3 | 84.0 | 76.2 | 16,721.2 | 4,910.4 | 2.4 | 2100 |
| 2 °C (n = 81) | Median | 50.9 | 34.2 | 4.8 | 59.4 | 58.4 | 62.8 | 7,977.4 | −2,377.8 | 1.5 | 2073 |
| | Mean | 51.4 | 34.3 | 6.3 | 59.4 | 56.0 | 61.2 | 8,732.4 | −2,462.1 | 1.5 | 2075 |
| | Min | 38.4 | 3.5 | 0.4 | 25.8 | 22.9 | 36.1 | 0 | −17,152.1 | 0.8 | 2050 |
| | Max | 77.0 | 63.6 | 16.6 | 93.7 | 88.7 | 88.7 | 18,778.6 | 6,039.5 | 2.2 | 2100 |
| >2 °C (n = 19) | Median | 53.2 | 28.7 | 3.4 | 63.7 | 58.4 | 75.4 | 11,260.4 | −962.1 | 1.3 | 2088 |
| | Mean | 52.6 | 31.2 | 7.3 | 61.5 | 53.3 | 73.0 | 10,505.6 | −993.9 | 1.3 | 2088 |
| | Min | 41.8 | 18.3 | 0.2 | 33.5 | 31.8 | 42.5 | 3,721.0 | −4,884.2 | 0.7 | 2066 |
| | Max | 66.7 | 51.7 | 23.4 | 78.7 | 72.3 | 108.1 | 16,355.9 | 5,410.9 | 2.9 | 2100 |
| All (n = 177) | Median | 48.5 | 32.7 | 4.8 | 62.2 | 55.2 | 56.6 | 7,481.3 | −2,503.8 | 1.6 | 2064 |
| | Mean | 49.5 | 33.8 | 6.2 | 60.0 | 53.9 | 56.8 | 8,048.3 | −2,406.6 | 1.5 | 2067 |
| | Min | 34.5 | 3.5 | 0 | 25.8 | 20.3 | 24.5 | 0 | −17,152.1 | 0.7 | 2037 |
| | Max | 79.9 | 63.6 | 23.4 | 93.7 | 88.7 | 108.1 | 18,778.6 | 6,039.5 | 2.9 | 2100 |
| | F-stat | 7.6 | 0.6 | 0.9 | 0.2 | 0.8 | 51.1 | 10.8 | 2.1 | 6.2 | 105.0 |
| | p-value | <0.001 | 0.539 | 0.394 | 0.808 | 0.472 | <0.001 | <0.001 | 0.121 | 0.002 | <0.001 |

Median, mean, minimum, and maximum for each scenario group are shown for each variable. In the final two rows, the F-statistic from a one-way ANOVA test is shown for the "all scenarios" group, indicating the magnitude of variation between warming group means for each variable across all 177 scenarios. Statistical significance of the F-statistic is indicated by p-values (bottom row). p-values <0.05 indicate that the variation between warming groups is statistically significant).

frequently show substantial negative emissions from land use in the Latin America region but the bulk of residual emissions occurring in other regions (Fig. 3b). Although the magnitude of negative emissions is not strongly related to the composition of the energy system, those scenarios with greater quantities of negative emissions from BECCS seem to also have greater levels of final energy demand and lower shares of non-biomass renewables (e.g., solar, wind, hydro; Fig. 4e). In contrast, the scenarios with greater negative emissions from land use (e.g., afforestation; represented by orange color in Fig. 4f) also have higher final energy demand, but have higher shares of non-biomass renewables (Fig. 4f). This reflects a logical trade-off in the availability of bioenergy and land-based carbon storage and suggests that the balance in IAMs outputs is being influenced by the level of future energy demand. However, it should be noted that prior studies have found that the value of negative emissions from BECCS will be more important than the value of generated electricity[54,55].

Finally, the relationships between energy use, GDP, and likely warming amount show that energy use is often limited in net-zero scenarios, especially for scenarios that limit warming to a greater extent (Fig. 1a). The median final energy consumption in global net-zero scenarios is 521 (range 227-857) EJ, compared to 418 EJ in 2019[56]. Given that global population is expected to reach nearly 9.5 billion by 2064 (median net-zero year) in SSP2[57], if per capita energy use remains constant at ~55 GJ/person, total final energy consumption will approach 523 EJ in 2064 – approximately equal to the net-zero scenario level. If instead per capita energy use continues to increase by about 0.16 GJ/person per year, as it did from 1971-2018 on average[56,58], total final energy consumption will approach 588 EJ in 2064 – 67 EJ above the net-zero scenario level. So, in order to limit final energy use to ~521 EJ in the median net-zero year, mean global per-capita energy use would have to remain nearly constant.

The process-based IAMs considered here have proven extraordinarily useful for articulating the overall shape of long-term mitigation pathways at a macro-regional to a global scale, but they are also limited in many ways that might influence our understanding of net-zero on a more detailed level. For example, because IAMs are designed to focus on larger-scale trends, they tend to have lower technological, temporal, and spatial resolutions compared with detailed energy system models[59,60] and do not consider the broad range of societal dynamics and political economy factors that can drive national emissions reduction strategies. Their strength in comprehensiveness is therefore balanced by limits to the detail in which they can represent regional or technological details that may be very relevant for actual strategy making, particularly with regard to rapid and disruptive technological change (e.g., management of electricity grids with high penetration of variable renewables, electric cars, greater digitalization, and hydrogen utilization pathways in heavy industry). Some studies have shown that because of this lower spatiotemporal detail, IAMs may be underestimating the role of variable renewables such as solar PV[60,61]. Furthermore, in this study we do not explicitly consider the detailed aspects of agriculture, forestry and other land use (AFOLU) sector and non-$CO_2$ emissions; however, these aspects are accounted for in the IAM frameworks themselves, which consistently include the linkages and tradeoffs between AFOLU and non-$CO_2$ emissions. The global full-economy representation provided by IAMs in this context makes them important tools in understanding pathways to net-zero greenhouse gas emissions balance as foreseen in Article 4 of the Paris Agreement. For all of these reasons, the net-zero scenarios we analyze here certainly do not reflect many of the details that will characterize net-zero emissions energy systems in the real world, but IAMs nonetheless remain critical bridges between more detailed energy systems models and long-term projections of climate change.

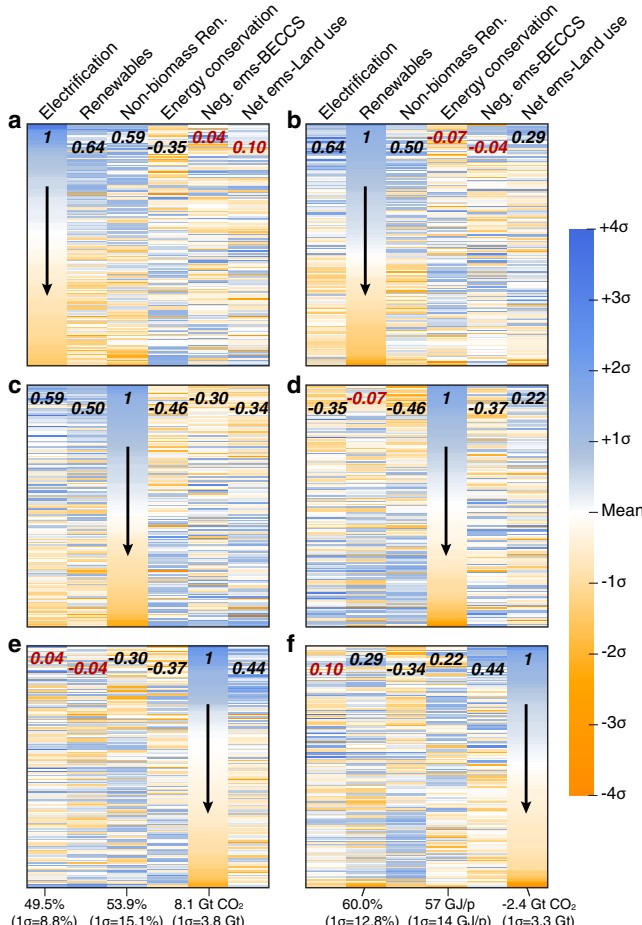

**Fig. 4 Relative characteristics of scenarios in the global net-zero year.**
Panels show parameter standard deviations for scenarios (rows) sorted by
(**a**) electrification, (**b**) renewables share, (**c**) non-biomass renewables
share, (**d**) energy conservation, (**e**) negative emissions from BECCS, and
(**f**) net land-use emissions. "Electrification" is the share of final energy
consumed as electricity. "Renewables" is the share of primary energy
supplied by biomass, solar, wind, hydroelectricity, and geothermal. "Non-
biomass ren." is the share of renewable energy sources provided by sources
other than biomass. "Energy conservation" here reflects the inverse of final
energy per capita, such that warmer colors indicate higher levels of energy
consumption. "Negative ems-BECCS" is the total amount of negative
emissions from bioenergy with carbon capture and storage. "Net ems-land
use" is the net amount of global CO2 emissions related to land use. Mean
and standard deviation for parameters are shown below each column, and
pairwise correlation coefficients (r) are shown in bold at the top of each
column. Black r-values are statistically significant ($p < 0.05$), while red
r-values are not.

In the time since the SR1.5 database was released, increased
efforts have been made to improve the model representation of
key technologies, such as carbon-neutral liquid fuels, long-term
storage of variable renewable energy, and negative emissions
strategies. Given that these results show liquid fuels remaining
prevalent and negative emissions strategies becoming increasingly
important in the existing net-zero scenarios, such modeling
improvements will be important to monitor going forward. The
relationship between higher residual emissions and correspond-
ing higher amounts of negative emissions in warmer scenarios
points toward reducing residual emissions as a target for policy
improvement, since negative emissions strategies are required to
offset any amount of residual emissions at net-zero. Reliance on

massive amounts of future negative emissions poses a substantial
risk, given that there is still considerable uncertainty surrounding
the feasibility of negative emissions technologies at such large
scales[15,35]. Policies that support carbon-neutral fuels and tech-
nologies now would in turn reduce future reliance on large
quantities of negative emissions to avoid harmful levels of
warming. Our findings thus represent an opportunity to assess
emerging net-zero emissions policies and energy trends in the
context of the longer-term global goal of limiting climate change.

## Methods
**Data source.** All of the model scenarios analyzed as part of this study were
obtained from the public 1.5 °C Scenario Database (the SR1.5 database), hosted by
the International Institute for Applied Systems Analysis (IIASA) through a process
facilitated by the Integrated Assessment Modelling Consortium (doi: 10.5281/
zenodo.3363345 | url: data.ene.iiasa.ac.at/iamc-1.5c-explorer). The model outputs
in the database were generated by the various Integrated Assessment Models
(IAMs) listed in Supplementary Table S1, and compiled by the Integrated
Assessment Modeling Consortium (IAMC)[31,32]. The full scenario set was curated
as part of the IPCC Special Report of Global Warming of 1.5 °C, Chapter 2 on
mitigation pathways and details of the models and scenarios are detailed in the
Technical Annex of the Chapter. The processes are described in more detail by
Huppmann et al.[31,32]. In this paper we use version r2.0 of the all regions dataset.
The 177 scenarios we assess here were produced by 7 main models (with 16
individual model variations), and thus are not truly independent of each other
since each IAM has its own assumptions built into the model framework.

While an updated scenario database is being developed for the upcoming IPCC
Sixth Assessment Report (AR6), our analysis is specifically about the characteristic
of the net-zero energy system at the point of net-zero, and not the pathway up to
that point. The broader insights of net-zero energy system characteristics gained
from our analysis are thus valuable and we expect they won't differ significantly in
subsequent analyses of the next generation of (AR6) scenarios. Moreover, although
recent developments in the power sector, e.g. renewables, have been faster than
expected, the observed values for 2019–2020 are still within the range of the
SR1.5 scenarios. For example, in 2020, approximately 2.9 EJ was generated from
solar electricity[62] and the SR1.5 scenario outputs for Secondary Energy|Electricity|
Solar in 2020 range from 0.2–6.6 EJ, with a median value of 1.8 EJ and a mean value
of 2.4 EJ. For wind energy, approximately 5.9 EJ was generated in 2020[62], and the
SR1.5 scenario outputs for Secondary Energy|Electricity|Wind in 2020 range from
1.0–23.6 EJ, with a median value of 7.4 EJ and a mean value of 6.9 EJ.

IAMs have a long and sometimes controversial history in their efforts to
characterize emissions pathways with the aim of mitigating climate change. The
IAMs here are primarily what would be considered as complex "process-based"
IAMs, as opposed to simpler "cost-benefit" IAMs that primarily simulate climate-
economy relationships to estimate the social cost of carbon[63].

They use a variety of over-arching modelling methods including linear
programming, partial- and computable general equilibrium, and recursive-dynamic
formulations. The models used tend to represent macro-economic regions,
comprising large countries and trading blocs, ranging from a few to tens of regions
with inter-regional trade of commodities, such as fuels and biomass. This regional
information was aggregated in the IPCC SR1.5 process to a common 5-region
definition (as above) to facilitate comparison. Temporal resolution is typically at 5
or 10-year timesteps, which is good for determining the levels of investments
required, whilst abstractions need to be made to ensure that reliability of electricity
systems remains plausible, such as ensuring that enough flexible reserve is available
to meet peak electricity demands.

Scenarios representing climate policy tend to be implemented using carbon
budget constraints that limit the cumulative carbon emissions over a period such
that warming does not pass the desired level, e.g. 2 °C. Further scenario-related
constraints may limit a wide range of parameters, such as technological options
and shares, rates of change and diffusion etc.

The IAMs whose scenarios we assess here do not include feedbacks from
climate impacts and damages, despite the fact that some studies have shown these
could be substantial[64,65]. Rather the models are designed to inform mitigation
efforts and have relatively simplistic representations of the Earth system[65]. Some
IAMs are beginning to include feedbacks between, for example, temperature
changes and energy use[66], and more ambitious efforts are underway that will
incorporate human energy, food and water systems into robust Earth system
models[67,68].

**Filtering and analysis of scenarios.** Our analysis includes only scenarios that
reach net-zero CO₂ emissions by the end of this century (year 2100). We define the
net-zero emissions year for each scenario (i.e., the x-axis in Fig. 1b) as the first year
that net global CO₂ emissions were equal to or less than zero. Because each model
produces parameter outputs at 5 or 10 year time steps, we interpolated annual data
using second-order polynomials.

We only consider CO₂ and not CH₄ or N₂O for several reasons. First, many of the
current net-zero policy targets are for net-zero CO₂ specifically[7]. Results from this

analysis will therefore be relevant to those policies in the context of net-zero $CO_2$. Second, entirely eliminating $CH_4$ or $N_2O$ emissions will entail the development of new technologies, particularly for removing these gases from the atmosphere[69], such that there are not yet practicable pathways to net-zero for these gases[7]. Third, $N_2O$ is primarily related to agriculture, and our analysis is focused on the energy system.

The scenarios are categorized into 6 regions (global and the five world regions defined in the SR1.5 database) and 3 consolidated levels of end-of-century global warming, based on the wider set determined in the IPCC report:

- 1.5 °C, which includes "below 1.5 °C," "1.5 °C return with low overshoot," "1.5 °C return with high overshoot";
- 2 °C, which includes "lower 2.0 °C" and "higher 2.0 °C," and;
- >2 °C, which corresponds to the category "above 2.0 °C". These scenarios have >50% likelihood of exceeding global mean temperature change of 2.0 °C by 2100, with no set upper bound of temperature change.

These global warming outcomes are primarily characterized by the "likely" (>50% chance) of reaching the specified temperature level by 2100. Further sub-categories of "overshoot" scenarios, based on the peak-warming and then return to a stabilization temperature help identify between scenarios that rely on substantial amounts of net-negative emissions.

The output variables for IAMs in the SR1.5 database are not entirely consistent; some models have extensive lists of outputs and regional and sectoral breakdowns, while others have comparatively few outputs and are missing some variables altogether. Our analysis therefore relies only on those IAM scenarios that include all output variables required for our analysis (177 out of 202 total net-zero emissions scenarios from the SR1.5 database; see Supplementary Table S1). Our interest in including as many scenarios as possible had to be balanced against our interest in exploring more detailed geographical and technological characteristics. Our analysis used the following 7 output variables: (1) $CO_2$ emissions (total net, energy and industrial processes net, AFOLU net), (2) Population, (3) GDP (PPP), (4) Primary energy, direct equivalent (total, fossil, nuclear, solar, wind, hydro, biomass), (5) Carbon Sequestration through BECCS, (6) Carbon price, and (7) Final energy (total and share from electricity). Residual $CO_2$ emissions were calculated by adding the residual emissions from energy and industrial processes (and, if applicable, the residual AFOLU emissions) to the amount of carbon sequestration from BECCS (since BECCS is used to offset residual emissions) in the net-zero year via the following equations:

If 'Emissions|$CO_2$|Energy and Industrial Processes' is *positive* at net-zero:

- 'Emissions|$CO_2$|Residual Fossil' = 'Emissions|$CO_2$|Energy and Industrial Processes' + 'Carbon Sequestration|CCS|Biomass'
  If 'Emissions|$CO_2$|Energy and Industrial Processes' is *negative* at net-zero:
- 'Emissions|$CO_2$|Residual Fossil' = 'Emissions|$CO_2$|Energy and Industrial Processes' + 'Carbon Sequestration|CCS|Biomass' + 'Emissions|$CO_2$|AFOLU'

All processing and analysis was done in JupyterLab (version 1.2.6). Code is available via GitHub: https://doi.org/10.5281/zenodo.5501623[70]

**Additional context for policymakers.** Around the world, countries and jurisdictions are adopting energy policies that mandate high levels of renewable or zero-carbon electricity in the next few decades[8,9]. For example, in the U.S., 14 states (California, Colorado, Hawaii, Maine, Maryland, Massachusetts, Nevada, New Mexico, New Jersey, New York, Oregon, Vermont, Virginia, and Washington) have laws requiring that >50% of electricity come from renewables such as wind, solar and biomass (but often excluding large-scale hydropower). Such goals are consistent with our analysis of net-zero scenarios generated by IAMs; renewables (including hydro) account for >50% of all primary energy in 74% of the net-zero scenarios. However, many places have pledged or mandated 100% renewable electricity and/or 100% net-zero emissions economy-wide by 2050, including the proposed EU Climate Law, and laws or government orders in the U.S. states of Hawaii, New York, Washington and California. Although details of these plans vary, it is noteworthy that very few of the net-zero scenarios reflect these goals at the macro-region level. This is due to the way that sources and sinks, from energy and land-use sectors, and between $CO_2$ and non-$CO_2$ sources, are optimized over much larger spatial extents including the influence of inter-regional trade, rather than the aforementioned policies that are enacted at state- and country-level. For example, the share of primary energy derived from renewables in the first year of net-zero or net-negative emissions is <80% in all but 2 of the 177 scenarios (Fig. 1c). Similarly, emissions in the OECD and EU region remain net-positive in more than half of the net-zero scenarios (pink points in Fig. 2d). Thus, we advise caution when interpreting these results, to note that the aforementioned zero-carbon energy policies are not necessarily over-ambitious or inconsistent with global and macro-regional IAM scenarios, because other nearby places and regions (e.g., Middle East and Africa), are likely to still be net-positive at the point at which global $CO_2$ emissions hit net-zero (Fig. 2d).

## Data availability
All of the model scenarios analyzed as part of this study were obtained from the public 1.5 °C Scenario Database (the SR1.5 database), hosted by the International Institute for Applied Systems Analysis (IIASA) through a process facilitated by the Integrated Assessment Modelling Consortium (https://doi.org/10.5281/zenodo.3363345 | url: data.ene.iiasa.ac.at/iamc-1.5-explorer).

## Code availability
All processing and analysis was done in JupyterLab (version 1.2.6). Code is available via GitHub: https://doi.org/10.5281/zenodo.5501623[70]

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

## Acknowledgements

The authors are grateful to Jinhyuk E. Kim for help in pre-processing scenario data. J.D. and S.J.D. acknowledge support from the U.S. National Science Foundation (INFEWS grant EAR 1639318). U.S. National Science Foundation (INFEWS grant EAR 1639318).

## Author contributions

S.J.D. and J.D. conceived the study. J.D. performed the analyses, with support from G.L., E.B., J.B. and S.J.D. J.D. and S.J.D. led the writing with input from I.A., J.B., G.L., E.B. and L.C.

## Competing interests
The authors declare no competing interests.
