## [Peer Review File · Nature Communications]

Energy systems in net-zero CO2 scenariosREVIEWER COMMENTS

Reviewer #1 (Remarks to the Author):

This is an interesting paper that analyzes the energy systems in all the modeling scenarios used for the 2018 IPCC report on limiting warming to 1.5°C. Specifically, the authors are interested in describing the characteristics of the global energy system when these systems are net-zero CO₂ emissions. While the paper makes a number of interesting observations about this data set and what it might teach us about energy futures, I have a number of questions and there are a number of ways in which the paper might be revised to improve its impact.

While the authors state that the dataset they are analyzing represents 177 net-zero scenarios, these are produced using a much smaller number of models so, in that sense, are not truly independent. The authors should discuss how many models were used to produce these inputs and also the fact that IAM results mirror each other to a large extent. They are produced by different groups in different countries but these groups work together and embed similar assumptions into their models.

The IPCC report was published in 2018 but the modeling scenarios were carried out in 2016, 2017. Given how quickly things are changing in the electric power space, are these results dated? Is it worth analyzing them?

In the abstract and in all their discussion, the percentages would be more useful if the authors described a benchmark. e.g., 60% renewables from a baseline today that is 30% coal:NG:nuclear and 10% renewables or whatever the numbers are.

The authors list the inequitable distribution of negative emissions technologies in the abstract, but that has been shown in a number of papers to be an output from IAMs over the past two years.

The statement on line 70 "The pathways to net-zero emissions..." is not clear to me. How do the authors expect that "pathways" and "characteristics" are different?

The paper lacks a real background/motivation.

The paper is missing references for a number of statements, such as the sentence that ends on line 128. Consider adding references for some of these statements that are not cited at present.

Line 141. This whole discussion about energy use per person would be more valuable if the authors contextualized these energy numbers in terms of what an average american, EU, or chinese resident uses today.

The paragraph that begins on Line 145 seems out of place.

Line 149. This strikes me as a chicken and egg logical structure. Is carbon pricing exogenous in all these modeling scenarios or is it a model output?

Line 177. Are these results from eastern europe and russia driven by climate or culture or cheap NG/coal?

Line 179. The equilibrium that is being described here (lots of positive emissions in some countries and negative emissions in other countries) seems like it would be hard to maintain for long.

Line 201. The authors should be clear, if they aren't in other parts of the paper, that none of these scenarios include DACCS. That is a major shortcoming in this data set since none of these modeling scenarios include negative emissions other than BECCS and/or afforestation.

Line 208. "The xx scenarios". should be fixed.

Line 230. This sentence does not appear to be in the right place and not in context.

Line 234. How does this impact the results?

Line 272. Do these scenarios include distributed generation such as rooftop solar?

Line 286. The discussion on around BECCS might need to be revised. Most analysis suggests that BECCS operations will primarily make their revenue primarily from carbon offsets not power production. In most of the scenarios the value of the negative emissions is much greater than the marginal price of power.

Reviewer #2 (Remarks to the Author):

This paper examines the energy systems of 177 IAM scenarios and analysis the general trend of the renewable energy sources, electrification, BECCS sequestration etc. The work is interesting and well written.

1. It would be nice to clarify the upper bound for the $>2^{\circ}\text{C}$ scenarios. It might be misleading for a free-to-emit situation, whereas they are still indicating an $>50\%$ likelihood or 2°C .

2. In this paper, the carbon price for 1.5°C , 2°C and $>2^{\circ}\text{C}$ scenarios are compared with the carbon sequestration amount from BECCS, but it is not clear how the carbon price is modelled in various IAMs. My understanding is mitigation cost can be measured from various cost metrics depends on model structure, e.g. 'Policy cost| consumption loss'/'Policy Cost|Area under MAC Curve'. And 'Area under MAC Curve' is more suitable for comparison with the BECCS sequestration.

3. Due to the difference in the modelling process, it is debatable whether the carbon price can be comparable among models. As can be seem from Figure 1 (f), higher carbon price is not indicating a lower temperature, and 1.5°C is showing less demand for BECCS in general.

4. Figure 2 (b) share from non-biomass renewables: should the biomass be at the upper part?

5. Typo Page 6 line 176: Gt CO2

6. In 'ADVANCE_2020_1.5C-2100, AIM/CGE 2.0', the 'Carbon Sequestration| CCS| Biomass' value is negative at the beginning while turns to be positive after 2050. The value in other scenario is positive through out this century. How do you treat this variable in your analysis?

Reviewer #3 (Remarks to the Author):

The paper provides a well-designed and informative picture of the socio-economic and energy conditions characterizing net-zero emission futures. First, the authors identify net-zero scenarios by querying the database constructed for the IPCC Special Report of Global Warming of 1.5°C . In particular, they select those scenarios that reach net-zero global emissions by 2100: 202 scenarios are found, of which 177 are used for the analysis. Second, they extract the quantitative outputs related to energy use, energy sources, residual emissions, electrification, gross domestic product and climate policy for each retained scenario and each available Integrated Assessment Model. While the literature has widely discussed pathways towards net-zero futures, the authors analyze the features of

the global energy system at a specific point in time, i.e., the year in which net-zero emissions are reached. Finally, they provide a series of statistics that describe how key variables distribute and correlate across scenarios at the time of net-zero emissions.

General comments/questions:

The analysis is based on descriptive evidence, which the authors use to build a number of points. While these are interesting and well organised, I have issues at finding the key innovative step of the paper. For example, insights on the share of renewables in primary energy and the sizable role of solid/liquid/gaseous fuels in deep decarbonisation scenarios are well visible in the Special Report on 1.5 Warming, as well as in other studies (which perhaps the authors cite: Luderer et al. 2018 NCC and Rogeli et al. 2018 NCC). Contrarily, my feeling is that the sources of regional heterogeneity in residual emissions and in carbon sequestration, and how they relate to the global energy system are still to be fully understood. The present paper provides relevant hints in such direction. At the moment, I feel the exposition of results is not satisfactory as it mixes known features of the 1.5 scenarios with intriguing novel trade-offs. My suggestion is to concentrate the analysis on the key trade-offs that emerge from the assessment of energy system at net-zero, and better explain them. I had hard times at getting the key message(s) from the paper.

Relatedly, I have some concerns about the statistical significance of the results that are presented. The authors accurately describe their findings making use of (i) differences in mean/median values across groups of scenarios (Fig. 1, 2 and 3), and (ii) unconditional correlations (Fig. 1, 2 and 4). However, they do not show nor discuss whether these differences and correlations are statistically significant. Sometimes the authors discuss associations by visual inspection of scatter plots (e.g. Lines 145-154). My feeling is that some more information/analyses can be useful to assess the degree of uncertainty behind the results.

In addition, I believe the analysis of Fig. 4 is relevant, but some more effort should be devoted at explaining the mechanisms behind the associations, rather than simply describing them (e.g. Lines 285-292).

Before moving to specific comments, there are two choices that I feel the authors should discuss further (maybe in the SI). First, the authors seem considering CO₂ emissions and not all GHGs. The choice is reasonable, but I wonder why, and whether results are robust if they had considered total emissions. Second, can the authors detail why in the analyses of Fig. 1 and 2 they considered negative emissions, but just from BECCS?. The debate on how to mix negative emissions technologies (NETs) for 1.5 futures is open (see Rueda et al. 2021, Glob. Env. Change); though the representation of NETs in IAMs is not wide, are there regional trade-offs emerging from the net-zero scenarios the authors focus on?

Specific comments:

Line 151. I think this is the first time the word "BECCS" appears. I suggest expanding the acronym.

Lines 191-192. The authors indicate that burden-sharing towards decarbonization among less-developed countries is not well balanced. I believe the authors should either provide some additional comments or remove the sentence. What would be a sound criterion to define a well-balanced effort? For example, some economies might be structurally more difficult to decarbonize than others given the same climate policy.

Line 208. I guess "xx" is a typo.

Line 365-366. While it is relevant to remark that impacts/damages are usually not accounted for in

the assessment of mitigation pathways, some additional references and/or explanations are needed here. I suggest the authors to discuss, perhaps in the SI, whether and how impacts are accounted for in the various scenarios they use. As climate damages might highly influence the patterns of decarbonization, the sentence at lines 365-366 is rather blurred and confusing at the moment.

Lines 240-262. I have some troubles at finding the rationale for a long paragraph that (i) enumerate a variety of policy initiatives for deep decarbonisation and (ii) discusses the inability of IAMs to model the necessary granularity that would be needed to evaluate/contextualise regional policies. It is known that IAMs cannot account for specific country/regional aimed or legislated policies. This is not what IAMs are designed for and I feel the whole paragraph adds little to the findings of the authors. I suggest shortening such discussion and, eventually, anticipating it in the introduction or postponing it to the point where other limitations of IAMs are discussed (e.g., pag. 9).

Lines 273-275. The finding that electricity is less prevalent in scenarios with lower warming is interesting, but poorly explained/investigated. The reason behind the result is not clear. Does it just depend on the role of liquid and solid fuels?

Lines 329-330. The policy implication that the authors derive is not clear to me. I would rephrase and explain with some more detail.

Suggestions:

One key step of the paper is to identify the year at which the system reaches net-zero emissions. Fig. 1b shows that the net-zero year is highly heterogenous across scenarios and does not really depend on when emissions peak. My feeling is that scenarios divide – roughly speaking – into “early” and “late” net-zero groups. At lines 101-102 the authors notice that the point in which net-zero emissions occur might affect the socio-economic and energy features of the system. My question is: does the trade-offs that characterize the global energy system differ between earlier or later net-zero decarbonizations?

Another concern I have regards a missing element in the analysis: investments. In particular, I wonder whether accounting for the (cumulative) investments in energy supply and energy efficiency – which are available in the SR1.5 database - can affect the results. The authors show, in Fig. 2, that the energy systems at the time of net-zero emissions are quite different across regions, reflecting heterogenous burden-sharing in emission mitigation and sequestration at the net-zero year. I wonder whether and how the picture changes considering the investments in the energy system across different regions. Is heterogeneity driven by different investment patterns? If not, what are the drivers?

RESPONSE TO REFEREES

We thank the editor and the reviewers for the thoughtful and valuable comments. We have made substantial revisions in response, and believe that the manuscript is considerably improved. Our revisions are detailed in the point-by-point responses below. The comments from the reviewers are in *blue italic* font and, our responses are in black regular font, and text in the revised manuscript are in *black italic* font.

Reviewer #1:

This is an interesting paper that analyzes the energy systems in all the modeling scenarios used for the 2018 IPCC report on limiting warming to 1.5°C. Specifically, the authors are interested in describing the characteristics of the global energy system when these systems are net-zero CO2 emissions. While the paper makes a number of interesting observations about this data set and what it might teach us about energy futures, I have a number of questions and there are a number of ways in which the paper might be revised to improve its impact.

Response: We thank Reviewer #1 for the positive summary of our work and for the thoughtful and constructive comments below.

While the authors state that the dataset they are analyzing represents 177 net-zero scenarios, these are produced using a much smaller number of models so, in that sense, are not truly independent. The authors should discuss how many models were used to produce these inputs and also the fact that IAM results mirror each other to a large extent. They are produced by different groups in different countries but these groups work together and embed similar assumptions into their models.

Response: As the Reviewer points out, the 177 scenarios we analyze were produced by 7 main models (with 16 individual model variations), and even those 7 are not entirely independent. This is a real limitation of IAM scenarios that in the context of the SR1.5 was discussed in both Rogelj et al. (2018a) and Huppmann et al. (2018). These discussions highlight that the scenario ensemble should not be treated as a statistical sample or be used to estimate scenario likelihood. Nonetheless, Section 2.3 of SR1.5 Chapter 2 states that “the combined body of scenarios can be assessed to identify salient features of pathways in line with a specific climate goal across a wide range of models” (Rogelj et al. 2018a), and similar methods have also been used for multi-model scenario comparison studies in recent years, including Luderer et al. (2018), Rogelj et al. (2018b), and Ghambir et al. (2019).

We have therefore revised the introductory text to make clear that the scenarios are not independent, that each is equally plausible given model constraints, and that the purpose of our analysis is not to determine likely futures but to identify ranges, characteristics, and tradeoffs in the net-zero energy systems represented in the database (L84-92).

1. Huppmann, D., Rogelj, J., Kriegler, E. *et al.* A new scenario resource for integrated 1.5 °C research. *Nature Clim Change* **8**, 1027–1030 (2018).
2. Gambhir, A., Rogelj, J., Luderer, G., Few, S. & Napp, T. Energy system changes in 1.5 °C, well below 2 °C and 2 °C scenarios. *Energy Strategy Reviews*, doi:10.1016/j.esr.2018.12.006. (2019).
3. Luderer, G. *et al.* Residual fossil CO2 emissions in 1.5–2 °C pathways. *Nature Climate Change* **8**, 626–633 (2018).
4. Rogelj, J., D. Shindell, K. Jiang, *et al.* Mitigation Pathways Compatible with 1.5°C in the Context of Sustainable Development. In: *Global Warming of 1.5°C. An IPCC Special Report on the impacts of global warming of 1.5°C above pre-industrial levels and related global greenhouse gas emission pathways, in the context of strengthening the global response to the threat of climate change, sustainable development, and efforts to eradicate poverty.* (2018a).
5. Rogelj, J. *et al.* Scenarios towards limiting global mean temperature increase below 1.5 °C. *Nature Climate Change* **8**, 325–332, doi:10.1038/s41558-018-0091-3 (2018b).

The IPCC report was published in 2018 but the modeling scenarios were carried out in 2016, 2017. Given how quickly things are changing in the electric power space, are these results dated? Is it worth analyzing them?

Response: Although a number of new scenarios have been produced and analyzed for the IPCC AR6, they are unfortunately not available for analysis and publication yet. However, our author group includes several of the lead and contributing AR6 authors, and we can attest that review of the newer model scenarios leads to conclusions that are broadly consistent with the SR1.5 scenarios we analyze.

Recent developments in power sector, e.g. renewables, have been faster than expected, but are still within range of the scenario set. For example, in 2020, approximately 2.94 EJ was generated from solar electricity (IEA Global Energy Review 2021), and the SR1.5 scenario outputs for Secondary Energy|Electricity|Solar range from 0.17-6.6 EJ, with a median value of 1.76 EJ and a mean value of 2.4 EJ. For wind energy, approximately 5.88 EJ was generated in 2020 (IEA Global Energy Review 2021), and the SR1.5 scenario outputs for Secondary Energy|Electricity|Wind range from 0.97-23.6 EJ, with a median value of 7.4 EJ and a mean value of 6.9 EJ.

Furthermore, we stress that the analysis is specifically about the characteristic of the net-zero energy system at the point of net-zero, and not the pathway up to that point. Thus we think the broader insights of net-zero energy system characteristics gained from our analysis are valuable and won't differ significantly in subsequent analyses of the next generation of (AR6) scenarios.

1. IEA (2021), *Global Energy Review 2021*, IEA, Paris <https://www.iea.org/reports/global-energy-review-2021>

In the abstract and in all their discussion, the percentages would be more useful if the authors described a benchmark. e.g., 60% renewables from a baseline today that is 30% coal:NG:nuclear and 10% renewables or whatever the numbers are.

Response: Thank you for this suggestion. We have added current values to the abstract for improved context, and we have emphasized in the discussion that the figures include modern reference values where applicable.

The authors list the inequitable distribution of negative emissions technologies in the abstract, but that has been shown in a number of papers to be an output from IAMs over the past two years.

Response: We have tried to reference the relevant prior work on this topic. To our knowledge, although studies may have explored regional differences in decarbonization scenarios, no one has highlighted this result in the context of global net-zero emissions. In response to the Reviewer's comment, we have added several new references (Honegger & Reiner 2018; van Soest et al. 2021) that have addressed the more general point of international distribution of future negative emissions (L215-218). If there are other relevant papers we've missed, we apologize and would appreciate the Reviewer pointing them out.

1. Honegger, M. & Reiner, D. The political economy of negative emissions technologies: consequences for international policy design. *Climate Policy*, 18:3, 306-321 (2018).
2. van Soest, H.L., den Elzen, M.G.J. & van Vuuren, D.P. Net-zero emission targets for major emitting countries consistent with the Paris Agreement. *Nat Commun* 12, 2140 (2021).

The statement on line 70 "The pathways to net-zero emissions..." is not clear to me. How do the authors expect that "pathways" and "characteristics" are different?

Response: We apologize that the distinction between pathways and characteristics was not well-explained in the original text. By comparing many scenarios at the point of net-zero emissions and identifying their common features and tradeoffs, we focus on the targeted end-state of energy systems

and deemphasize the timing and incremental steps toward that goal. We recognize that this is a subtle difference, but we think it is useful in this case to collapse the temporal dimension, set aside the near-term trajectories of these scenarios, evaluate *de novo* the characteristics of plausible net-zero systems as they are portrayed by the models. This is because it is entirely possible that systems could evolve in ways and at rates not represented in any of the scenarios but reach the same composition at net-zero emissions. We have revised the main text to clarify these points (L67-70).

The paper lacks a real background/motivation.

Response: Given this feedback and the Reviewer’s previous comments, we revised the introductory text to explain our motivation and the gap we believe this paper fills (L67-70, L84-92). In particular, most of the modeling literature about net-zero emissions is regional and/or sectorally-narrow in scope and most of the IAM literature to date has been focused on decarbonization or climate-stabilization pathways and not the end goal of net-zero emissions. Our analysis therefore shows what global net-zero emissions systems look like in IAMs, the breadth of technological composition and energy sources, persistent regional disparities, and trade-offs we think are relevant and important to policymakers.

The paper is missing references for a number of statements, such as the sentence that ends on line 128. Consider adding references for some of these statements that are not cited at present.

Response: We appreciate the comment and have added more references throughout, including in support of the indicated sentence (now ending on L133).

Line 141. This whole discussion about energy use per person would be more valuable if the authors contextualized these energy numbers in terms of what an average american, EU, or chinese resident uses today.

Response: Great suggestion. We’ve added a sentence that contextualizes per capita energy use by comparing the scenario values at net-zero to what the average American, EU, and Chinese residents use today (L154-155).

The paragraph that begins on Line 145 seems out of place.

Response: We had organized the paragraphs to follow the order of the main text figures, but agree that the first sentence of the paragraph the Reviewer is referring to did not provide adequate transition from the previous paragraph and therefore was a bit jarring. We have edited this in the main text (now beginning on L157). If the Reviewer has a suggestion for where the paragraph might fit better in the text that is different than its current placement, we would be happy to move it.

Line 149. This strikes me as a chicken and egg logical structure. Is carbon pricing exogenous in all these modeling scenarios or is it a model output?

Response: Carbon pricing in the SR1.5 scenarios is a model output (calculated endogenously), representing the marginal cost of abatement required to achieve an exogenous climate goal (Rogelj et al. 2018a; Guivarch and Rogelj 2017; Riahi et al. 2017). We have edited this section in the main text to clarify the interpretation of carbon price given this model structure. We thank the reviewer for pointing out the confusing logic in the sentence beginning on Line 163 which was due to a typographical error that we’ve now corrected.

“It is important to note that carbon prices in the SR1.5 scenarios are endogenous “shadow” carbon prices that reflect the marginal cost of abatement, and thus do not directly reflect the impact of explicit (exogenous) carbon pricing such as a carbon tax or cap-and-trade system^{28,33,34}. The relationship between BECCS and carbon price should therefore be interpreted as the impact of marginal abatement cost on BECCS deployment. The lack of a clear relationship between the two

does not necessarily mean that marginal abatement cost is inconsequential for the magnitude of negative emissions, but rather indicates that other dynamics relating to technology availability and costs may be the main drivers of BECCS deployment.” (L160-168)

1. Rogelj, J., D. Shindell, K. Jiang, *et al.* Mitigation Pathways Compatible with 1.5°C in the Context of Sustainable Development. In: *Global Warming of 1.5°C. An IPCC Special Report on the impacts of global warming of 1.5°C above pre-industrial levels and related global greenhouse gas emission pathways, in the context of strengthening the global response to the threat of climate change, sustainable development, and efforts to eradicate poverty.* (2018a).
2. Guivarch, C. & Rogelj, J. *Carbon price variations in 2°C scenarios explored.* Carbon Pricing Leadership Coalition, USA (2017).
3. Riahi, K. *et al.* The Shared Socioeconomic Pathways and their energy, land use, and greenhouse gas emissions implications: An overview. *Global Environmental Change* **42**, 153-168 (2017).

Line 177. Are these results from eastern europe and russia driven by climate or culture or cheap NG/coal?

Response: Thank you for this question. We have attempted to clarify the mechanisms behind regional differences in the main text. In summary, since these process-based IAMs use cost-effectiveness analysis (CEA; Rogelj *et al.*, 2018a) which minimizes the total mitigation costs of reaching a specified climate goal, all associations between output variables reflect what is cheapest.

“Since the process-based IAMs considered here use cost-effectiveness analysis (CEA)²⁸, which minimizes the total mitigation costs of reaching a specified climate goal, all associations between output variables are essentially a reflection of what is cheapest. For example, in a scenario where substantial residual emissions remain at net-zero and are offset by correspondingly large amounts of negative emissions, reducing gross emissions to zero must have been more expensive than continuing to emit and offsetting with negative emissions. The most cost-effective outputs for scenarios are also based on the assumptions of individual models, including the availability and cost of technologies.” (L261-L269)

1. Rogelj, J., D. Shindell, K. Jiang, *et al.* Mitigation Pathways Compatible with 1.5°C in the Context of Sustainable Development. In: *Global Warming of 1.5°C. An IPCC Special Report on the impacts of global warming of 1.5°C above pre-industrial levels and related global greenhouse gas emission pathways, in the context of strengthening the global response to the threat of climate change, sustainable development, and efforts to eradicate poverty.* (2018a).

Line 179. The equilibrium that is being described here (lots of positive emissions in some countries and negative emissions in other countries) seems like it would be hard to maintain for long.

Response: We agree with the Reviewer that this positive-negative regional emissions equilibrium would be hard to maintain and have added additional discussion about this in the main text (L215-222).

Line 201. The authors should be clear, if they aren't in other parts of the paper, that none of these scenarios include DACCS. That is a major shortcoming in this data set since none of these modeling scenarios include negative emissions other than BECCS and/or afforestation.

Response: Out of the 177 net-zero scenarios we analyzed, only 6 scenarios included DACCS, all from the same model (REMIND 1.7). We thought this was too few to analyze and report on, and agree with the Reviewer that the inclusion of more negative emissions technologies in future scenarios will strengthen the comprehensiveness of this dataset. However, despite the lack of a range of NETs in these scenarios, the magnitude of BECCS is suggestive of the magnitude of total negative emissions that may be required at net-zero.

Line 208. “The xx scenarios”. should be fixed.

Response: Thank you for catching this. The “xx scenarios” in Lines 238-239 should read “40 scenarios” and has been corrected in the main text.

Line 230. This sentence does not appear to be in the right place and not in context.

Response: We have edited the indicated sentence to improve the transition between paragraphs and give better context:

“To further explore this relationship between negative emissions and other parameters, the underlying structure of the IAMs is also important to consider: some of the SR1.5 models are partial equilibrium models (e.g. POLES ADVANCE) while others are general equilibrium (e.g. AIM-CGE 2.0 and 2.1) or hybrid models (e.g. MESSAGE-GLOBIOM 1.0) that link the two²⁶. Additionally, certain scenarios have conditions that limit the amount or type of negative emissions technology used, such as EMF33_1.5C_limbio, which sets a limit of 100 EJ/year for the amount of bioenergy from BECCS, cellulosic fuels, and hydrogen²⁶.” (L270-276)

Line 234. How does this impact the results?

Response: To show how underlying model structure/assumptions impacts the scenario output, we have added a figure to the Supplementary Information (Figure S7, included below for convenience) showing the ranges of values for each model for residual emissions, non-biomass renewable energy share, and electrification, organized by warming level category (reproduced below for convenience). The previous Figure S7 is now Figure S8. We have also added a section in the text describing how the model differences shown in Figures S7 and S8 might impact the results (beginning on L276).

“Supplementary Figure S7 shows the scenario ranges for residual emissions, non-biomass renewable energy share, and electrification for each model. These ranges demonstrate how the structure and assumptions of individual models affect these scenario outputs: for example, GCAM scenarios tend to have systematically higher residual emissions and lower amounts of non-biomass renewable energy and electrification than those of other models (Figure S7). Such model differences are visible in comparing individual scenarios, but the output ranges are more sensitive to the scenario constraints than on the models (Figure S8).” (L276-283)

Supplementary Figure S7 | Model comparison for parameters in global net-zero year. Residual CO₂ emissions per capita (top row), the share of renewable primary energy from non-biomass sources (middle row), and the share of final energy as electricity (bottom row) are shown for 1.5°C (left column), <2.0°C (middle column), and >2.0°C (right column) scenarios, broken out by model. In each case, the boxes show the range from 25th to 75th percentiles, and the whiskers indicate the 5th and 95th percentiles where applicable (some models did not have enough scenario points to show 5th and 95th percentiles, since any points outside the box range were outliers). Lines and circles within the boxes denote the median and mean values, respectively. 15 models ran 1.5° scenarios, 16 models ran <2.0° scenarios, and 9 models ran >2.0° scenarios that reach net-zero emissions. For ease of comparison, box colors are consistent for each model throughout the figure (each model is always the same color across plots).

Line 272. Do these scenarios include distributed generation such as rooftop solar?

Response: The only net-zero scenarios that explicitly include distributed PV in the SR1.5 documentation are run by GCAM 4.2; the rest only report generation under the more general category of “Solar PV.”

Line 286. The discussion on around BECCS might need to be revised. Most analysis suggests that BECCS operations will primarily make their revenue primarily from carbon offsets not power production. In most of the scenarios the value of the negative emissions is much greater than the marginal price of power.

Response: We have added a sentence to the revised text reflecting the Reviewer’s point.

“However, it should be noted that prior studies have found that the value of negative emissions from BECCS will be more important than the value of generated electricity (Fajardy et al. 2021; Muratori et al. 2016).” (L314-315)

1. Fajardy, M. *et al.* The economics of bioenergy with carbon capture and storage (BECCS) deployment in a 1.5°C or 2°C world. *Global Environmental Change* **68**, 102262 (2021).
2. Muratori, M. *et al.* Global economic consequences of deploying bioenergy with carbon capture and storage (BECCS). *Environ. Res. Lett.* **11**, 095004 (2016).

Author Responses to Reviewer #2:

Reviewer #2 (Remarks to the Author):

This paper examines the energy systems of 177 IAM scenarios and analysis the general trend of the renewable energy sources, electrification, BECCS sequestration etc. The work is interesting and well written.

Response: We thank Reviewer #2 for the careful review of our manuscript and positive appraisal.

1. It would be nice to clarify the upper bound for the >2°C scenarios. It might be misleading for a free-to-emit situation, whereas they are still indicating an >50% likelihood or 2°C.

Response: A good point. We’ve revised to clarify the upper bound of likely temperatures for the >2°C scenarios both in the main text (L93-96) and the Methods section (L424-426):

“Figure 1 shows the relationships among global energy and socioeconomic variables in the year of global net-zero emissions, broken out by the level of projected global warming. These categories include overshoot scenarios that return to the specified amount of warming by end-of-century (see Methods).” (L93-96)

“These scenarios have >50% likelihood of exceeding global mean temperature change of 2.0°C by 2100, with no set upper bound of temperature change.” (L424-426)

2. In this paper, the carbon price for 1.5°C, 2°C and >2°C scenarios are compared with the carbon sequestration amount from BECCS, but it is not clear how the carbon price is modelled in various IAMs. My understanding is mitigation cost can be measured from various cost metrics depends on model structure, e.g. ‘Policy cost| consumption loss’/‘Policy Cost|Area under MAC Curve’. And ‘Area under MAC Curve’ is more suitable for comparison with the BECCS sequestration.

Response: As the Reviewer is probably aware, carbon pricing in the SR1.5 scenarios is a model output (endogenous), representing the marginal cost of abatement required to achieve an exogenous climate goal (Rogelj et al. 2018a; Guivarch and Rogelj 2017; Riahi et al. 2017). We have edited this section in the main text to clarify the interpretation of carbon price given this (new text shown below for ease of reference). We did not consider the ‘Policy Cost|Area under MAC Curve’ variable because only scenarios run by the IMAGE and POLES models had this output. We thus had to balance our interest in including as many scenarios as possible against the limits of the variables provided in the dataset; since the carbon price output was available for many more models/scenarios, we decided to use this variable instead.

“It is important to note that carbon prices in the SR1.5 scenarios are endogenous “shadow” carbon prices that reflect the marginal cost of abatement, and thus do not directly reflect the impact of explicit (exogenous) carbon pricing such as a carbon tax or cap-and-trade system^{28,33,34}. The relationship between BECCS and carbon price should therefore be interpreted as the impact of

marginal abatement cost on BECCS deployment. The lack of a clear relationship between the two does not necessarily mean that marginal abatement cost is inconsequential for the magnitude of negative emissions, but rather indicates that other dynamics relating to technology availability and costs may be the main drivers of BECCS deployment.” (L160-168)

1. Rogelj, J., D. Shindell, K. Jiang, *et al.* Mitigation Pathways Compatible with 1.5°C in the Context of Sustainable Development. In: *Global Warming of 1.5°C. An IPCC Special Report on the impacts of global warming of 1.5°C above pre-industrial levels and related global greenhouse gas emission pathways, in the context of strengthening the global response to the threat of climate change, sustainable development, and efforts to eradicate poverty.* (2018a).
2. Guivarch, C. & Rogelj, J. *Carbon price variations in 2°C scenarios explored.* Carbon Pricing Leadership Coalition, USA (2017).
3. Riahi, K. *et al.* The Shared Socioeconomic Pathways and their energy, land use, and greenhouse gas emissions implications: An overview. *Global Environmental Change* **42**, 153-168 (2017).

3. Due to the difference in the modelling process, it is debatable whether the carbon price can be comparable among models. As can be seen from Figure 1 (f), higher carbon price is not indicating a lower temperature, and 1.5°C is showing less demand for BECCS in general.

Response: A number of previous studies have compared carbon price across different IAMs (Rogelj *et al.* 2018a; Rogelj *et al.* 2018b; Luderer *et al.* 2018), justifying the comparison by the fact that the carbon price reflects the marginal cost of abatement (cost effective method, or CEA as described in SR1.5 Chapter 2) in the SR1.5 scenarios for all of the models (both recursive dynamic framework and intertemporal optimization framework models) (Rogelj *et al.* 2018a; Guivarch and Rogelj 2017).

In light of the Reviewer’s comment, we have added some additional explanation (L160-163) which we hope will help readers to evaluate the comparison. However, if the Reviewer is convinced that the comparison should not be presented, we could remove it without losing much.

1. Rogelj, J., D. Shindell, K. Jiang, *et al.* Mitigation Pathways Compatible with 1.5°C in the Context of Sustainable Development. In: *Global Warming of 1.5°C. An IPCC Special Report on the impacts of global warming of 1.5°C above pre-industrial levels and related global greenhouse gas emission pathways, in the context of strengthening the global response to the threat of climate change, sustainable development, and efforts to eradicate poverty.* (2018a).
2. Rogelj, J. *et al.* Scenarios towards limiting global mean temperature increase below 1.5 °C. *Nature Climate Change* **8**, 325-332, doi:10.1038/s41558-018-0091-3 (2018b).
3. Luderer, G. *et al.* Residual fossil CO₂ emissions in 1.5–2 °C pathways. *Nature Climate Change* **8**, 626-633 (2018).
4. Guivarch, C. & Rogelj, J. *Carbon price variations in 2°C scenarios explored.* Carbon Pricing Leadership Coalition, USA (2017).

4. Figure 2 (b) share from non-biomass renewables: should the biomass be at the upper part?

Response: We thank the Reviewer for pointing out our unclear labeling (which applied to both figure 2b and 1c). The arrows pointing down for “biomass” and up for “solar, wind, hydro” were intended to emphasize that points below the 50% share of renewable primary energy from non-biomass renewables use more biomass than non-biomass sources, and points above 50% on that same axis use more non-biomass sources (e.g. solar, wind, and/or hydro) than biomass. We have clarified the labeling by changing “biomass” to “more biomass” and “solar, wind, hydro” to “more solar, wind, hydro.”

5. Typo Page 6 line 176: Gt CO₂

Response: We have corrected the mistake, thanks.

6. In ‘ADVANCE_2020_1.5C-2100, AIM/CGE 2.0’, the ‘Carbon Sequestration/ CCS/ Biomass’ value

is negative at the beginning while turns to be positive after 2050. The value in other scenario is positive throughout this century. How do you treat this variable in your analysis?

Response: For our analysis, we use version r2.0 of the all regions dataset (which includes global values). The values for ‘Carbon Sequestration|CCS|Biomass’ are all either zero or positive for these scenarios over the period 2010-2050 (including the scenario pointed out by the Reviewer is referencing (a plot produced by the IAMC 1.5°C Scenario Explorer is shown below). We therefore treat this variable in our analysis as the magnitude of negative emissions from BECCS (e.g., a value of 50Mt for Carbon Sequestration|CCS|Biomass is treated as 50Mt of negative CO₂ emissions, equivalent to -50Mt of CO₂ in terms of its impact on net emissions).

Author Responses to Reviewer #3:

Reviewer #3 (Remarks to the Author):

The paper provides a well-designed and informative picture of the socio-economic and energy conditions characterizing net-zero emission futures. First, the authors identify net-zero scenarios by querying the database constructed for the IPCC Special Report of Global Warming of 1.5°C. In particular, they select those scenarios that reach net-zero global emissions by 2100: 202 scenarios are found, of which 177 are used for the analysis. Second, they extract the quantitative outputs related to energy use, energy sources, residual emissions, electrification, gross domestic product and climate policy for each retained scenario and each available Integrated Assessment Model. While the literature has widely discussed pathways towards net-zero futures, the authors analyze the features of the global energy system at a specific point in time, i.e., the year in which net-zero emissions are reached. Finally, they provide a series of statistics that describe how key variables distribute and correlate across scenarios at the time of net-zero emissions.

Response: We appreciate the Reviewer’s positive summary and constructive comments.

General comments/questions:

The analysis is based on descriptive evidence, which the authors use to build a number of points. While these are interesting and well organised, I have issues at finding the key innovative step of the paper. For example, insights on the share of renewables in primary energy and the sizable role of solid/liquid/gaseous fuels in deep decarbonisation scenarios are well visible in the Special Report on 1.5 Warming, as well as in other studies (which perhaps the authors cite: Luderer et al. 2018 NCC and Rogeli et al. 2018 NCC). Contrarily, my feeling is that the sources of regional heterogeneity in residual emissions and in carbon sequestration, and how they relate to the global energy system are still to be fully understood. The present paper provides relevant hints in such direction. At the moment, I feel the exposition of results is not satisfactory as it mixes known features of the 1.5

scenarios with intriguing novel trade-offs. My suggestion is to concentrate the analysis on the key trade-offs that emerge from the assessment of energy system at net-zero, and better explain them. I had hard times at getting the key message(s) from the paper.

Response: In response to the Reviewer's concerns about the novelty of our study—which were consistent with Reviewer #1's concerns about the paper's motivation and background, we have made substantial revisions to the introductory text and have expanded discussion of the points the Reviewer suggests were most interesting.

In particular, most of the modeling literature about net-zero emissions is regional and/or sectorally-narrow in scope and most of the IAM literature to date has been focused on decarbonization or climate-stabilization pathways and not the end goal of net-zero emissions. Our analysis is focused on what global net-zero emissions systems look like in IAMs—a key gap in the current literature, including the breadth of technological composition and energy sources, as well as the persistent regional disparities and trade-offs mentioned as interesting by the reviewer.

Relatedly, I have some concerns about the statistical significance of the results that are presented. The authors accurately describe their findings making use of (i) differences in mean/median values across groups of scenarios (Fig. 1, 2 and 3), and (ii) unconditional correlations (Fig. 1, 2 and 4). However, they do not show nor discuss whether these differences and correlations are statistically significant. Sometimes the authors discuss associations by visual inspection of scatter plots (e.g. Lines 145-154). My feeling is that some more information/analyses can be useful to assess the degree of uncertainty behind the results.

Response: To address statistical significance of our results, we calculated p-values for each of the correlations shown in Figure 4 (which effectively summarizes the correlations described in Figures 1-3). We have updated Figure 4 to reflect the statistical significance of the Pearson correlation coefficients (r) shown in the figure: black correlation coefficients are statistically significant ($p < 0.05$), while red correlation coefficients are not statistically significant. These p-values are not stating any confidence interval about future of the energy system (which we cannot do given that the scenarios we analyze are not a representative sample of such futures; Huppmann et al., 2018), but reflect our confidence in the robustness of correlations between variable outputs. We have also updated the text with p-values where we describe correlations that are not shown in Figure 4, e.g. L203-205.

In keeping with previous IAM studies (e.g., Luderer et al. 2018; Rogelj et al. 2018b; Riahi et al. 2017), we indicate uncertainty in the multi-model/scenario comparisons by ranges in the model outputs (e.g., Table 1 and the various box plots). In response to the Reviewer's comment, we have now revised the text to consistently report such ranges throughout.

1. Huppmann, D., Rogelj, J., Kriegler, E. *et al.* A new scenario resource for integrated 1.5 °C research. *Nature Clim Change* **8**, 1027–1030 (2018).
2. Rogelj, J. *et al.* Scenarios towards limiting global mean temperature increase below 1.5 °C. *Nature Climate Change* **8**, 325-332, doi:10.1038/s41558-018-0091-3 (2018b).
3. Luderer, G. *et al.* Residual fossil CO₂ emissions in 1.5–2 °C pathways. *Nature Climate Change* **8**, 626-633 (2018).
4. Riahi, K. *et al.* The Shared Socioeconomic Pathways and their energy, land use, and greenhouse gas emissions implications: An overview. *Global Environmental Change* **42**, 153-168 (2017).

In addition, I believe the analysis of Fig. 4 is relevant, but some more effort should be devoted at explaining the mechanisms behind the associations, rather than simply describing them (e.g. Lines 285-292).

Response: We agree that more discussion of the mechanisms behind these relationships was necessary, which is now included in Lines 261-269 of the main text (shown below). In summary, since these process-based IAMs use cost-effectiveness analysis (CEA; Rogelj et al., 2018a) which

minimizes the total mitigation costs of reaching a specified climate goal, all associations between output variables reflect what is cheapest.

“Since the process-based IAMs considered here use cost-effectiveness analysis (CEA)²⁸, which minimizes the total mitigation costs of reaching a specified climate goal, all associations between output variables are essentially a reflection of what is cheapest. For example, in a scenario where substantial residual emissions remain at net-zero and are offset by correspondingly large amounts of negative emissions, reducing gross emissions to zero must have been more expensive than continuing to emit and offsetting with negative emissions. The most cost-effective outputs for scenarios are also based on the assumptions of individual models, including the availability and cost of technologies.” (L261-L269)

1. Rogelj, J., D. Shindell, K. Jiang, *et al.* Mitigation Pathways Compatible with 1.5°C in the Context of Sustainable Development. In: *Global Warming of 1.5°C. An IPCC Special Report on the impacts of global warming of 1.5°C above pre-industrial levels and related global greenhouse gas emission pathways, in the context of strengthening the global response to the threat of climate change, sustainable development, and efforts to eradicate poverty.* (2018a).

*Before moving to specific comments, there are two choices that I feel the authors should discuss further (maybe in the SI). First, the authors seem considering CO₂ emissions and not all GHGs. The choice is reasonable, but I wonder why, and whether results are robust if they had considered total emissions. Second, can the authors detail why in the analyses of Fig. 1 and 2 they considered negative emissions, but just from BECCS?. The debate on how to mix negative emissions technologies (NETs) for 1.5 futures is open (see Rueda *et al.* 2021, *Glob. Env. Change*); though the representation of NETs in IAMs is not wide, are there regional trade-offs emerging from the net-zero scenarios the authors focus on?*

Response: We only consider CO₂ and not CH₄ or N₂O for several reasons. First, many of the current net-zero policy targets are for net-zero CO₂ specifically (Rogelj *et al.* 2021). Results from this analysis will therefore be relevant to those policies in the context of net-zero CO₂. Second, entirely eliminating CH₄ or N₂O emissions will entail development of new technologies, particularly for removing these gases from the atmosphere (Jackson *et al.* 2019), such that there are not yet practicable pathways to net-zero for these gases (Rogelj *et al.* 2021). Third, N₂O is primarily related to agriculture, and our analysis is focused on the energy system. Nonetheless, these reasons were not mentioned in the original text and should have been. We have added some explanation in the Methods section (L412-417).

We neglected negative emissions from land use both because it is not directly related to the energy system (except via BECCS) and because there are reporting inconsistencies in negative land use emissions in the scenario database. We do show net land use emissions in Figure 3, but this output could not be broken down any further into residual vs. negative emissions from land use.

1. Jackson, R.B., Solomon, E.I., Canadell, J.G. *et al.* Methane removal and atmospheric restoration. *Nat Sustain* **2**, 436–438 (2019).
2. Rogelj, J. *et al.* Net-zero emissions targets are vague: three ways to fix. *Nature* **591**, 365-368 (2021).

Specific comments:

Line 151. I think this is the first time the word “BECCS” appears. I suggest expanding the acronym.

Response: Thank you for pointing this out. We define BECCS where it first appears in Line 159.

Lines 191-192. The authors indicate that burden-sharing towards decarbonization among less-developed countries is not well balanced. I believe the authors should either provide some additional comments or remove the sentence. What would be a sound criterion to define a well-balanced effort? For example, some economies might be structurally more difficult to decarbonize than others given the same climate policy.

Response: Thank you for pointing this out. We have revised and expanded the discussion of burden-sharing in the context of equity-based emissions allocation to Lines 215-222 of the main text:

“This supports recent research on regional and country-level negative emissions distributions in the context of regional net-zero emissions^{35,36} and indicates that burden-sharing between currently less-developed regions may not be well-balanced in IAM outputs when global emissions reach net-zero. While there are many different approaches to defining a well-balanced mitigation effort³⁷, burden-sharing approaches that consider equity as a key component are vital for meeting sustainable development goals³⁸. Analysis of the SR1.5 scenarios in the context of equitable emissions/negative emissions allocation and sustainable development warrants further research.” (L215-222)

Line 208. I guess “xx” is a typo.

Response: Thank you for catching this. The “xx scenarios” in Lines 238-239 should read “40 scenarios” and has now been corrected.

Line 365-366. While it is relevant to remark that impacts/damages are usually not accounted for in the assessment of mitigation pathways, some additional references and/or explanations are needed here. I suggest the authors to discuss, perhaps in the SI, whether and how impacts are accounted for in the various scenarios they use. As climate damages might highly influence the patterns of decarbonization, the sentence at lines 365-366 is rather blurred and confusing at the moment.

Response: Based on this suggestion, we have revised the discussion about climate damages in the SR1.5 scenarios in the Methods section.

“The IAMs whose scenarios we assess here do not include feedbacks from climate impacts and damages, despite the fact that some studies have shown these could be substantial^{57,58}. Rather the models are designed to inform mitigation efforts and have relatively simplistic representations of the Earth system⁵⁸. Some IAMs are beginning to include feedbacks between, for example, temperature changes and energy use⁵⁹, but more ambitious efforts are underway that will incorporate human energy, food and water systems into robust Earth system models^{60,61}. (L401-406)

1. Calvin, K. and Bond-Lamberty, B. Integrated human-earth system modeling—state of the science and future directions. *Environ. Res. Lett.* **13**, 063006 (2018).
2. Calvin, K. *et al.* GCAM v5.1: representing the linkages between energy, water, land, climate, and economic systems. *Geosci. Model Dev.* **12**, 677-698 (2019a).
3. Calvin, K. *et al.* Characteristics of human-climate feedbacks differ at different radiative forcing levels. *Global and Planetary Change* **180**, 126-135 (2019b).
4. Collins, W. D. *et al.* The integrated Earth system model version 1: formulation and functionality. *Geosci. Model Dev.*, **8**, 2203–2219 (2015).
5. Woodard, D.L. *et al.* Economic carbon cycle feedbacks may offset additional warming from natural feedbacks. *Proceedings of the National Academy of Sciences* **116**(3), 759-764 (2018).

Lines 240-262. I have some troubles at finding the rationale for a long paragraph that (i) enumerate a variety of policy initiatives for deep decarbonisation and (ii) discusses the inability of IAMs to model the necessary granularity that would be needed to evaluate/contextualise regional policies. It is known that IAMs cannot account for specific country/regional aimed or legislated policies. This is not what IAMs are designed for and I feel the whole paragraph adds little to the findings of the authors. I suggest shortening such discussion and, eventually, anticipating it in the introduction or postponing it to the point where other limitations of IAMs are discussed (e.g., pag. 9).

Response: As suggested, we have moved this paragraph to the Methods section where other context/limitations of IAMs are discussed (L450-470).

Lines 273-275. The finding that electricity is less prevalent in scenarios with lower warming is interesting, but poorly explained/investigated. The reason behind the result is not clear. Does it just depend on the role of liquid and solid fuels?

Response: Thank you for pointing this out. We have edited this section and added several sentences to better explain this finding and hope this adds clarity.

“Warmer scenarios may have slightly higher levels of electrification: median shares of 1.5 °C, 2 °C and >2 °C scenarios are 46% (range 35-80%), 51% (range 38-77%), and 53% (range 42-67%), respectively, perhaps because they afford greater time for end-uses to transition (Fig. 1e). This transition-time effect on amount of electrification is supported by Supplementary Figure S9, which shows that scenarios that reach net-zero after the median net-zero year (2064) tend to have higher amounts of electrification (Fig. S9e). Warming amount is also correlated to both net-zero year ($r = 0.73$, $p < 0.001$; Fig. 1b) and electrification ($r = 0.27$, $p < 0.001$) in the Fig. 1 global scenarios, which further supports the idea that warmer scenarios have slightly higher amounts of electrification because they reach net-zero emissions later, thus allowing more time for end-uses to transition and for costs to decline.” (L142-151)

Lines 329-330. The policy implication that the authors derive is not clear to me. I would rephrase and explain with some more detail.

Response: As suggested, we revised the explanation for this policy implication in Lines 351-356:

“The relationship between higher residual emissions and corresponding higher amounts of negative emissions by warming level points toward reducing residual emissions as a target for policy improvement, since negative emissions strategies are required to offset any amount of residual emissions at net-zero. Policies that support carbon-neutral fuels and technologies now would in turn reduce the amount we must rely on large amounts of negative emissions to avoid harmful levels of warming in the future.” (L351-356)

Suggestions:

One key step of the paper is to identify the year at which the system reaches net-zero emissions. Fig. 1b shows that the net-zero year is highly heterogenous across scenarios and does not really depend on when emissions peak. My feeling is that scenarios divide – roughly speaking – into “early” and “late” net-zero groups. At lines 101-102 the authors notice that the point in which net-zero emissions occur might affect the socio-economic and energy features of the system. My question is: does the trade-offs that characterize the global energy system differ between earlier or later net-zero decarbonizations?

Response: Thank you for this suggestion. In response, we have made a new figure (Figure S9) that looks at the same parameters as Figure 1 but colors the points according to whether they reach net-zero before (“Early”) or after (“Late”) the median net-zero year for all global scenarios (2064). Based on this figure, we also now discuss the impact of early vs. late net-zero years on electrification in Lines 145-151 of the main text:

“This transition-time effect on amount of electrification is supported by Supplementary Figure S9, which shows that scenarios that reach net-zero after the median net-zero year (2064) tend to have higher amounts of electrification (Fig. S9e). Warming amount is also correlated to both net-zero year ($r = 0.73$, $p < 0.001$; Fig. 1b) and electrification ($r = 0.27$, $p < 0.001$) in the Fig. 1 global scenarios, which further supports the idea that warmer scenarios have slightly higher amounts of electrification because they reach net-zero emissions later, thus allowing more time for end-uses to transition and for costs to decline.” (L145-151)

Supplementary Figure S9 | Early vs. Late net-zero scenarios. Global scenarios that reach net-zero emissions before or at the median net-zero year 2064 (“Early”, blue points) vs. after the median net-zero year (“Late”, orange points) show differences in energy use (a), emissions trajectory (b), energy sources (c), residual emissions (d), electrification (e), and policy (f). Points represent individual scenarios, with frequency of scenarios shown along each axis for Early and Late scenario groups. Colored dashed lines and values indicate medians for Early vs. Late scenario groups. Gray dashed lines indicate reference values for the year shown in gray.

Another concern I have regards a missing element in the analysis: investments. In particular, I wonder whether accounting for the (cumulative) investments in energy supply and energy efficiency – which are available in the SRI.5 database - can affect the results. The authors show, in Fig. 2, that the energy systems at the time of net-zero emissions are quite different across regions, reflecting heterogeneous burden-sharing in emission mitigation and sequestration at the net-zero year. I wonder whether and how the picture changes considering the investments in the energy system across different regions. Is heterogeneity driven by different investment patterns? If not, what are the drivers?

Response: We thank the reviewer for pointing this out and in response have another new figure (Figure S10) showing the relationship between cumulative investment, electrification, BECCS, and residual energy+industry emissions, and have added some discussion of impact of investments based on this figure in Lines 200-207 of the main text:

“These differences in residual and negative emissions across regions can be at least partially explained by differences in investment: Supplementary Figure S10 shows that cumulative investment in non-fossil electricity supply up to the global net-zero year is correlated with regional electrification ($r = 0.40$, $p < 0.001$), negative emissions from BECCS ($r = 0.45$, $p < 0.001$), and residual emissions from energy and industry ($r = 0.63$, $p < 0.001$) (Figure S10). Of course, investment is not the only cost-related driver of these regional characteristics, but it does appear to play a significant role in the smaller subset of scenarios that include investment output values.” (L200-207)

Supplementary Figure S10 | Regional investment vs. electrification, negative emissions from BECCS, and residual energy+industry emissions. Scenarios that reach net-zero emissions globally and have regional outputs for investment in non-fossil electricity supply ($n=20$ scenarios with all regions, for a total of 100 data points) show regional differences in share of final energy as electricity (top), negative emissions from BECCS (middle), and residual emissions from energy and industry (bottom). Points represent individual scenarios, with frequency of scenarios shown along each axis for each region (Asia = blue, Latin America = green, Middle East+Africa = orange, OECD+EU countries = pink, and Eastern Europe+Russia = purple). Colored dashed lines and values indicate medians for each region.

References

1. Calvin, K. and Bond-Lamberty, B. Integrated human-earth system modeling—state of the science and future directions. *Environ. Res. Lett.* **13**, 063006 (2018).
2. Calvin, K. *et al.* GCAM v5.1: representing the linkages between energy, water, land, climate, and economic systems. *Geosci. Model Dev.* **12**, 677-698 (2019a).
3. Calvin, K. *et al.* Characteristics of human-climate feedbacks differ at different radiative forcing levels. *Global and Planetary Change* **180**, 126-135 (2019b).
4. Collins, W. D. *et al.* The integrated Earth system model version 1: formulation and functionality. *Geosci. Model Dev.*, **8**, 2203–2219 (2015).
5. Fajardy, M. *et al.* The economics of bioenergy with carbon capture and storage (BECCS) deployment in a 1.5°C or 2°C world. *Global Environmental Change* **68**, 102262 (2021).
6. Gambhir, A., Rogelj, J., Luderer, G., Few, S. & Napp, T. Energy system changes in 1.5 °C, well below 2 °C and 2 °C scenarios. *Energy Strategy Reviews*, doi:10.1016/j.esr.2018.12.006. (2019).
7. Guivarch, C. & Rogelj, J. *Carbon price variations in 2°C scenarios explored*. Carbon Pricing Leadership Coalition , USA (2017).
8. Huppmann, D., Rogelj, J., Kriegler, E. *et al.* A new scenario resource for integrated 1.5 °C research. *Nature Clim Change* **8**, 1027–1030 (2018).
9. Honegger, M. & Reiner, D. The political economy of negative emissions technologies: consequences for international policy design. *Climate Policy*, 18:3, 306-321 (2018).
10. IEA (2021), *Global Energy Review 2021*, IEA, Paris <https://www.iea.org/reports/global-energy-review-2021>
11. Luderer, G. *et al.* Residual fossil CO₂ emissions in 1.5–2 °C pathways. *Nature Climate Change* **8**, 626-633 (2018).
12. Muratori, M. *et al.* Global economic consequences of deploying bioenergy with carbon capture and storage (BECCS). *Environ. Res. Lett.* **11**, 095004 (2016).
13. Riahi, K. *et al.* The Shared Socioeconomic Pathways and their energy, land use, and greenhouse gas emissions implications: An overview. *Global Environmental Change* **42**, 153-168 (2017).
14. Rogelj, J., D. Shindell, K. Jiang, *et al.* Mitigation Pathways Compatible with 1.5°C in the Context of Sustainable Development. In: *Global Warming of 1.5°C. An IPCC Special Report on the impacts of global warming of 1.5°C above pre-industrial levels and related global greenhouse gas emission pathways, in the context of strengthening the global response to the threat of climate change, sustainable development, and efforts to eradicate poverty.* (2018a).
15. Rogelj, J. *et al.* Scenarios towards limiting global mean temperature increase below 1.5 °C. *Nature Climate Change* **8**, 325-332, doi:10.1038/s41558-018-0091-3 (2018b).
16. Rogelj, J. *et al.* Net-zero emissions targets are vague: three ways to fix. *Nature* **591**, 365-368 (2021).
17. van Soest, H.L., den Elzen, M.G.J. & van Vuuren, D.P. Net-zero emission targets for major emitting countries consistent with the Paris Agreement. *Nat Commun* **12**, 2140 (2021).
18. Woodard, D.L. *et al.* Economic carbon cycle feedbacks may offset additional warming from natural feedbacks. *Proceedings of the National Academy of Sciences* **116**(3), 759-764 (2018).

REVIEWERS' COMMENTS

Reviewers #1 and #2 had no additional comments and were in support of publication.

Reviewer #3 (Remarks to the Author):

I would like to thank the authors for their replies and for having updated the paper in response to my comments. I still have some concerns, which I feel the authors should consider. I detail them in the attached Word file.